# On the origins of conductive pulse sensing inside a nanopore

Lauren S. Lastra[1], Y. M. Nuwan D. Y. Bandara [1], Michelle Nguyen[2], Nasim Farajpour[1] & Kevin J. Freedman[1✉]

Nanopore sensing is nearly synonymous with resistive pulse sensing due to the characteristic occlusion of ions during pore occupancy, particularly at high salt concentrations. Contrarily, conductive pulses are observed under low salt conditions wherein electroosmotic flow is significant. Most literature reports counterions as the dominant mechanism of conductive events (a molecule-centric theory). However, the counterion theory does not fit well with conductive events occurring via net neutral-charged protein translocation, prompting further investigation into translocation mechanics. Herein, we demonstrate theory and experiments underpinning the translocation mechanism (i.e., electroosmosis or electrophoresis), pulse direction (i.e., conductive or resistive) and shape (e.g., monophasic or biphasic) through fine control of chemical, physical, and electronic parameters. Results from these studies predict strong electroosmosis plays a role in driving DNA events and generating conductive events due to polarization effects (i.e., a pore-centric theory).

[1] Department of Bioengineering, University of California, Riverside, 900 University Avenue, Riverside, CA 92521, USA. [2] Department of Biology, University of California, Riverside, 900 University Avenue, Riverside, CA 92521, USA. ✉email: kfreedman@engr.ucr.edu

Since their first use as a biosensor, solid-state nanopores continue to explore new biophysical phenomena and have cemented their place in history as high-throughput, low-cost overhead, real-time, single-molecule resolution electrical read-out platforms. Although the translocation profiling of biochemically, biomedically, and pharmaceutically impactful new molecules and particles has gained tremendous traction in laboratories across the world, the high electrolyte concentration paradigm in which experiments are performed has been rather unchanged since the sensing inception of nanopores in 1996[1]. The attractiveness associated with high electrolyte solutions is largely due to the high signal-to-noise ratio (SNR), high electrophoretic throughput, and reliable generation of resistive pulses stemming from DNA transiently blocking ions (typically potassium and chloride). The physical principles in which DNA modulates the flow of ionic current within a nanopore have been studied extensively[2–4]. Although nanopore sensing is mostly associated with resistive pulse sensing due to transient ionic current perturbation by the molecule, the resistive nature of events is not consistent across all DNA translocation experiments[1,5,6]. In 2004, Chang et al.[7] reported on current-enhancing events at low electrolyte concentrations wherein the DNA-occupied pore conducted more ions compared to the DNA-free pore. Therefore, pulses generated through translocations can be categorized as either current-reducing (i.e., resistive event, RE), or current-enhancing (i.e., conductive event: CE).

As electrolyte concentration decreases, CEs are often observed in both planar membrane nanopores as well as conical nanopipettes, suggesting that CEs are not pore geometry specific[8–16]. It is also at this regime where electroosmotic flow (EOF) strengthens, sometimes leading to the translocation of molecules opposing electrophoretic flow (EPF). Although EOF and CEs often coincide, it is important to note that they are not mechanistically linked. For example, CEs are seen in nanopores where EOF is reduced to allow EPF-driven events[9]. Despite the large number of experiments describing CEs, the origins of CEs in the presence of low ionic strength have been elusive. The leading consensus is that the combination of additional counterions and frictional effects influence the production of CEs[3]. Specifically, CEs stem from the introduction of additional counterions by the charged DNA (i.e., $K^+$) within the nanopore is greater than the number of ions within the DNA-free pore[7]. Once electrolyte concentration decreases below ~0.02 M, mostly counterions are present within the pore, which explains the current enhancement[17,18]. Interestingly, at ~0.4 M, counterions are thought to precisely compensate for the DNA-occupied regions of the pore and yields no current modulation[19].

The results presented herein conflict with the conventional consensus and may be better explained by another potential theory; namely that current enhancement is due to a flux imbalance, which causes (1) charge density polarization and (2) voltage changes at the pore ($V_{pore}$). Indeed, the first report of nanopore sensing at asymmetric salt conditions suggested that $V_{pore}$ may be reduced and was used as an explanation for slower DNA translocations[20]. Perhaps the most convincing evidence, presented here, for the need of a new model lies with the fact that conductive events are observed for proteins at both symmetric low-salt conditions and asymmetric high-salt conditions. The heterogeneous surface charge of proteins would mean that counterions would be of mixed valency ($+e$, $−e$). Even if positive counterions were more prevalent on the surface of the protein, we would expect the current enhancement to be minimal. Instead, we found that the current enhancement is greater than that of DNA. The flux imbalance theory presented here does not depend on the analyte at all but rather is modeled using the steady state flux of ions through pore.

Asymmetric high-salt conditions, explored by Zhang et al.[21], also produced CEs and the authors used a multi-ion model composed of Nernst-Plank and Stokes equations to explain their observations. Namely, EOF enhancement in the space between the DNA and the pore is significantly higher than the ions blocked by DNA occupancy in the pore[21]. Our experimental observations with PEG (a natively neutral polymer that functions as a polycationic polymer through cation adsorption) cannot be explained through this model where CEs were seen with smaller diameter pores (Supplementary Information). Protein (i.e., holo form of human serum transferrin) translocation under a low ionic strength condition, yielded CEs as well (Supplementary Information). Thus, the intriguing question, why does ionic current increase during transient DNA and protein occupancy of a nanopore, remains under-examined and warrants further investigation. Since a cohesive theory for the nature of conducting events is still elusive, we studied the transport of DNA and protein within a nanopipette using various monovalent salts and under symmetric and asymmetric salt conditions.

A second fundamental question that remains debated in the literature is: can low-salt conditions promote EOF-driven DNA transport. Although it may seem obvious, electroosmotic dominant transport of DNA is hardly reported (first predicted in 2010[22]) and therefore, less known in the nanopore community[23,24]. On the other hand, electrophoretic transport of DNA through nanopores is well-reported and almost unanimously used mode of transport. While electroosmosis has seen widespread adoption in protein and glycan characterization, its use in DNA experiments is meager, largely due to the high linear charge density associated with DNA and (high) salt conditions typically used in experiments[25,26]. However, tuning of electroosmosis has been used to, for example, promote single file translocation, improve throughput, and tune translocation time[25,27,28]. To the best of our knowledge, no previous reports exist outlining the electroosmotic DNA transport through nanopipettes. Thus, herein, we characterized EOF-driven events (anti-electrophoretic, or anti-EPF) with Lambda DNA (λ-DNA)—the gold standard of the nanopore community to benchmark new developments due to its well-known physicochemical parameters—using quartz nanopipettes.

In summary of our findings, DNA CEs are extremely cation-, pore size-, and voltage-specific and potentially the result of an imbalance of ionic fluxes and leads to charge density polarization and a violation of net neutrality[29]. We utilize a Poisson–Nernst–Planck (PNP) model to describe the flux imbalance between cation and anions within a nanopore, which differs from the more traditional Nernst–Planck (NP) equations in how electro-neutrality and charge conservation is formulated. The PNP model more accurately describes the boundary layers (1–10 nm) at electrodes and charged surfaces[30]. For nanopores that are on the same order of magnitude as the boundary layers, the PNP equations are a more complete treatment of charged species transport. The net effect is that flux imbalances have the ability to change the space charge density and the voltage throughout the fluidic system. We will discuss the electrokinetic and hydrodynamic phenomena that affects event shapes such as counterion cloud, ion mobility, pore size, and electrolyte composition. This report elucidates some of the fundamental prerequisites for observing CEs when DNA translocates through a nanopore and paves the way for harnessing CE mechanisms for DNA sequencing and single molecule biophysical discoveries.

## Results and discussion

While most nanopore-based, single-molecule sensing is performed using planar membranes, which have a well-defined pore length (i.e., effective membrane thickness), nanopipettes have a

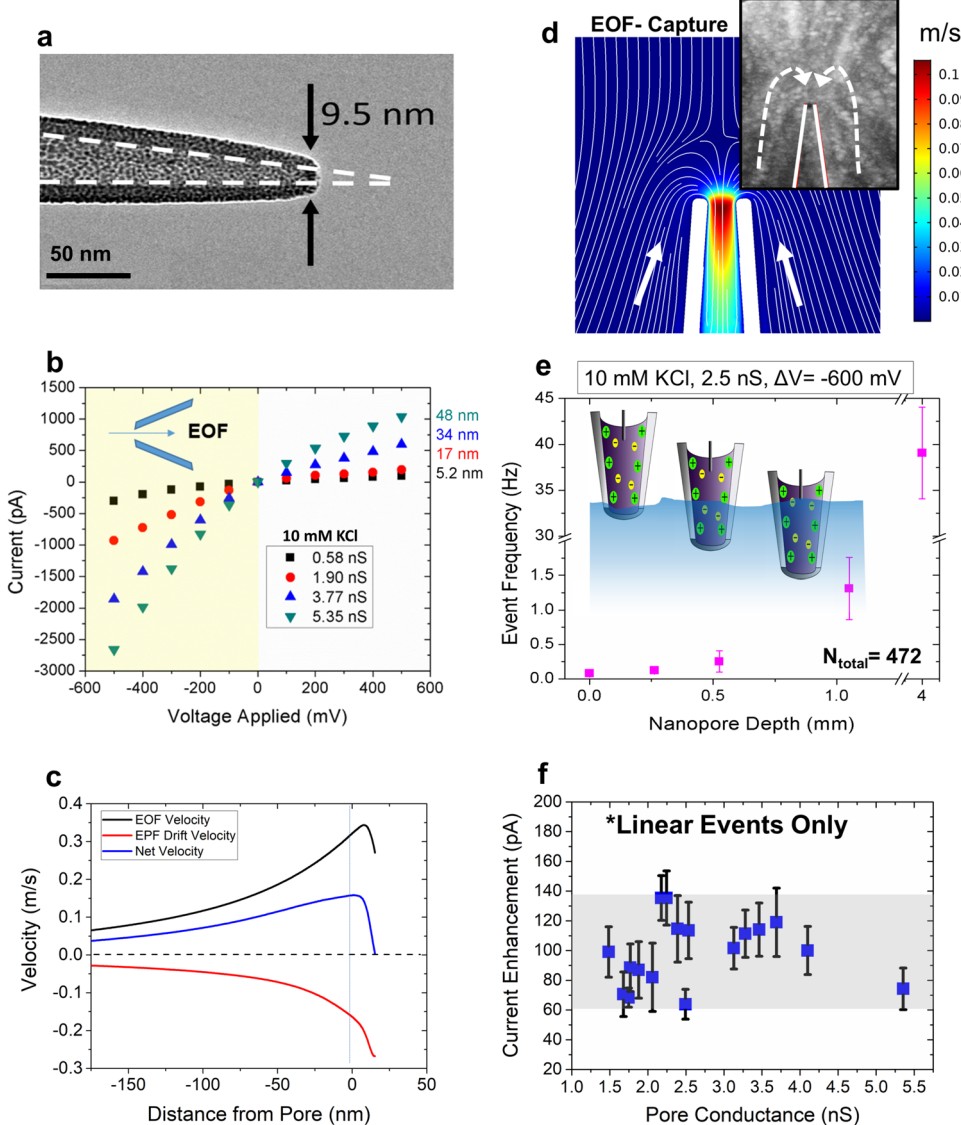

**Fig. 1 Experimental set-up and characterization of quartz nanopores. a** TEM of quartz nanopore; scale bar, 50 nm. **b** I–V curves pertaining to four differently sized nanopipette orifices. For pore size estimations, the linear portion at the negative voltages was used (yellow shaded region). The schematic within the I–V curves shows the directionality of EOF and EPF at negative voltages. **c** EOF, EPF drift, and the resulting net velocities of λ-DNA along the pore's axis of symmetry ($\mu = 3.2 \times 10^4$ cm/Vs). Distance from the pore is radial from the axis of symmetry. **d** Simulations of fluid flow velocities under low ionic strength conditions. White lines indicate fluid flow lines for a 20 nm pore at −600 mV voltage bias. Inset: YOYO-labeled DNA sample with an applied voltage of −700 mV to visualize the capture zone. **e** Event frequency with depth of the pipette inside the bath solution. Nanopore depth is synonymous with how deep the nanopore tip was submerged into the analyte-containing bath solution. Error bars show the standard deviation of each condition. **f** Linear DNA events from 17 pores were investigated for pore size dependence on current amplitude. We see that the enhancements fluctuate between 60 and 140 pA with no discernable trend. Errors bars represent the standard deviation of the current change.

gradual taper length (Fig. 1a) that increases the sensing region of the device[31]. We fabricated nanopipettes by laser pulling glass nanocapillaries, producing two identical quartz nanopores (see Methods for fabrication details). With this technique, <10 nm inner pore diameters can be achieved as shown in Fig. 1a. This process is fast, inexpensive, and does not require a clean-room environment[32]. The pore conductance (G) was evaluated using the linear slope of a current–voltage (I–V) curve (Fig. 1b) and thereafter used to estimate the size of the aperture using[33,34]:

$$d_i = \frac{4Gl}{\pi K d_b} \qquad (1)$$

where l is the length of the conical pore (taper length), K is the measured conductivity of the buffer, and $d_b$ is the diameter of the capillary (0.7 mm) at the beginning of the conical taper. The taper length was measured using an optical microscope. The G, measured by calculating the slope of the linear portion at the negative voltages, varied between 0.6 and 5.4 nS and the I–V curve showed ionic current rectification, which is consistent with the previous reports[35]. The tabulated G values yield pore diameters between 5 (±0.5) and 48 (±4) nm, respectively. The pore sizes were also occasionally confirmed using transmission electron microscopy (see Supplementary Information for further details).

**λ-DNA translocation in symmetric low-salt conditions**. After retrieving the $I–V$ information, translocation experiments with λ-DNA at a final concentration of 500 pM were performed in 10 mM Tris-EDTA buffered at pH ~7.4. We opted for very a low-salt concentration (i.e., 10 mM) to maximize the EOF while maintaining a high enough SNR for pulse extraction (see discussion in Supplementary Information for more details on SNR). The pH was maintained at the physiological pH, which renders the glass to be negatively charged ($\approx -(10–20)$ mC/m$^2$)[36] and therefore EOF and EPF to be opposing in the case of λ-DNA. For electroosmotic capture to take place, it should outweigh the electrophoretic force (provided the two are opposing rather than complementing) exerted on the DNA molecule by the applied voltage. For a molecule to translocate, it must first diffuse to the capture zone, drift to the pore opening and overcome electrostatic and free energy barriers (e.g., entropy). The shape and extent of the capture volume are exceedingly crucial as they would govern the transport dynamics of the device. It is known, when EPF dominates, the capture volume outside the nanopore assumes a nearly spherical shape surrounding the pore's orifice[20,37–40]. EOF, on the other hand, depends on the fluid flow profiles. According to the EOF streamlines, the capture volume adopts a shape confined along the sides of the pore[41]. There also lies a crossover concentration point in which EOF reverses direction, where EOF is generated along the glass surface and radiates away from the pore aperature[41]. Finite element analysis was performed to determine the fluid flow rate at different voltages (Fig. 1c). Herein, we adopted the operational configuration where the anode electrode is placed inside the pipette side and grounded electrode in the bath (under low-salt conditions). Since the glass surface is negatively charged at the operational pH, at negative applied voltages, the resultant fluid flow would be towards the taper region (i.e., from the bath to the tip). At positive biases, the fluid flow direction switches. In brevity, the simulation depicted in Fig. 1c was carried out in the following manner: Poisson–Nernst–Planck–Stokes equations were solved simultaneously to account for ionic species spatial concentrations, electrostatic forces on ions and convective forces on ions. EOF was imposed as a force on the surrounding liquid by integrating the spatial accumulation of ions into a volume force that acts on the liquid (boundary conditions can be found and simulations details can be found in the Supplementary Information). The fluid velocity acts as a moving frame of reference for the DNA and can be compared directly with the electrophoretic drift velocity imposed by the electric field. Electrophoretic drift velocity was calculated by extracting the electric field and multiplying by the electrophoretic mobility of DNA ($\mu = 3.2 \times 10^4$ cm/Vs)[42]. Simulated results shown in Fig. 1c indicate, under low ionic strength conditions, the EOF velocity is greater than the EPF drift velocity rendering the net velocity to be in the same direction as the EOF profile.

Given the inherent differences associated with capture volume shapes associated with EOF and EPF dominant mechanisms, the next step was to elucidate the entrance trajectory of DNA. To do this, λ-DNA was added to the bath and a negative voltage bias was applied to the other electrode to ensure if translocations were to happen (i.e., from the bath to the tip side; forward translocation direction), it would be caused by electroosmosis rather than by the conventional electrophoresis. The fluid flow profiles around pore-tip were simulated to further understand the EOF-driven capture of DNA. The simulated results are shown in Fig. 1d and indicate DNA proceeds to diffuse around the solution until it enters the EOF capture volume, where it is then transported through the pore. To reiterate, this transport is fundamentally possible when the EOF velocity is greater than the EPF drift velocity. Since DNA events occur anti-EPF, mapping the fluid motion is indicative of the capture zone. To experimentally validate the finite element analysis (Fig. 1d), λ-DNA was tagged with YOYO-1 and the nanopipette tip placed in the focal plane of a water immersion objective (Nikon, NA = 1.2). A stacked time series of images (acquired from a Princeton Instruments ProEM emCCD) allowed us to observe λ-DNA capture at −700 mV (Fig. 1d inset reveals that fluid motion along the sides of the pore is responsible for λ-DNA translocation).

Under high-salt conditions, DNA transport has been categorized to adopt a range of configurations, including linear, looped, partially folded, and knotted: reported with both planar nanopores[43–46] and nanocapillaries[33]. However, reports on the DNA conformations under EOF dominant transport are yet to be published. Thus, after confirming the mode of dominant transport, we looked at the conformations adopted by translocating DNA molecules. Realizing that the capture volume in EOF-driven translocations surrounds the outer walls of the nanopipette, we first optimized the throughput of the device by adjusting the pipette position with respect to the bath liquid surface as shown in Fig. 1e. The capture volume can be controlled by submerging varying lengths of the taper length inside the salt solution containing λ-DNA. The nanopore was suspended at 0, 0.26, 0.53, 1.1, and 4.0 mm below the electrolyte solution surface containing λ-DNA. For exact measurements, the nanopore was suspended from a linear stage actuator. Translocations were obtained for voltages between −100 and −1000 mV, in increments of 100 mV. Recording at −600 mV yielded the most consistent translocations without clogging the pore. Events were recorded at −600 mV and the $I–V$ relationship yielded a 2.5 nS pore. The capture rate was calculated at each depth (see Supplementary Information for capture rate calculation details). As nanopore depth increases, capture volume also increases, leading to higher event frequency with larger depth values. As more of the nanopore is exposed to the λ-DNA solution, the capture volume enlarges, leading to an increase in event frequency and corroborates the EOF capture mechanism more strongly than the electrophoretic capture mechanism. Finally, using a custom-coded MATLAB script, translocation conformations of DNA were examined, which revealed that DNA adopts the widely seen conformations: linear, partially folded, and fully folded (see Supplementary Information for further details). By solely selecting linear events, we were able to evaluate the relationship between CE amplitude and pore size; a relationship that may be hidden by multiple conformations of DNA. As seen in Fig. 1f, no observable trends were seen in CE amplitude with pore conductance (a proxy for pore size).

**λ-DNA translocation in asymmetric high-salt conditions**. Simulations performed in 2009 predicted that current enhancements could be seen at high ionic strength conditions[47] with small pore diameters (<2.2 nm) using hairpin DNA. In acknowledgment of that finding, we also show that CE phenomenon is not limited to low ionic strength conditions. We employed the usage of salt concentration gradients where the pipette was filled with 1 M KCl and the bath was filled with 4 M KCl. λ-DNA was either added to the pipette (Fig. 2b: case I) or bath (Fig. 2c: case II) and a voltage bias consistent with the conventional EPF-dominated transport was applied to the pipette. Note that EOF is deemed negligible under the high-salt conditions that these experiments operate. In case I, with an applied voltage of −600 mV, λ-DNA was driven outside the pore through EPF, resulting in CEs. This contradicts the conventional expectation of REs under high-salt conditions and CEs under low-salt conditions. On the contrary, in case II, with an applied voltage of +600 mV, λ-DNA was driven into the pipette resulting

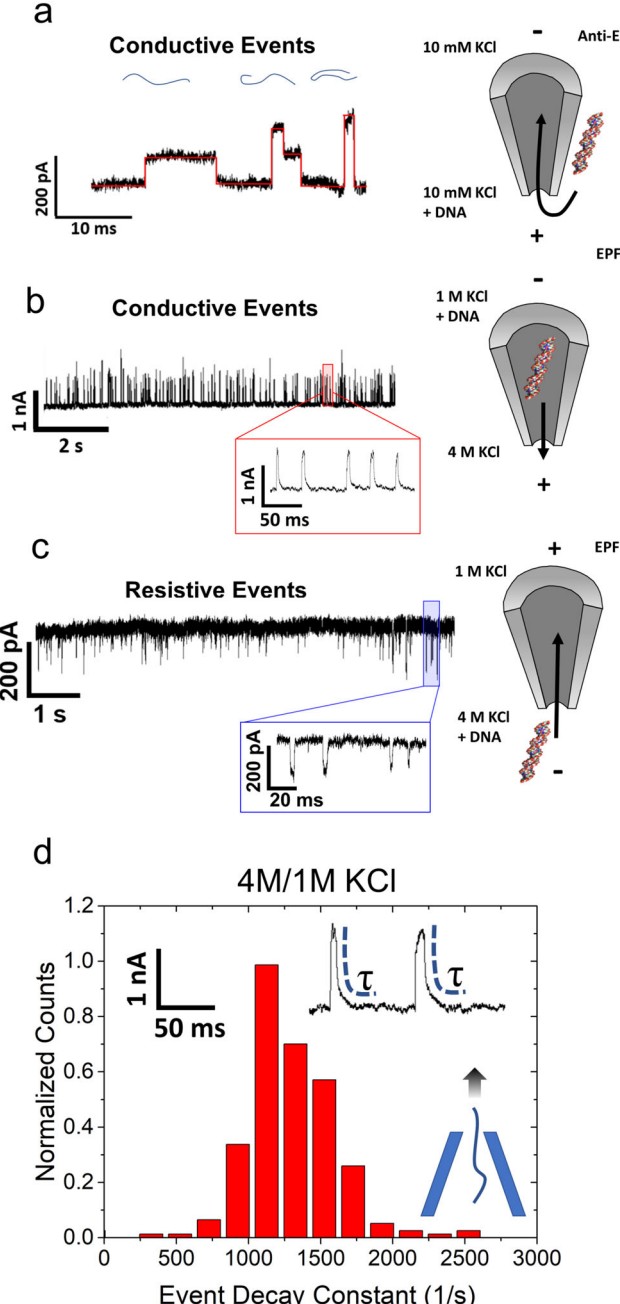

**Fig. 2 Event properties of DNA under low ionic strength and asymmetric salt conditions. a** Typical event structures observed with λ-DNA translocation experiments under low salt, symmetric conditions at −600 mV. The three events correspond to linear, partially folded, and fully folded λ-DNA from left to right (current traces in black overlaid with red lines). The corresponding blue lines show an example of each DNA configuration. The schematic on the right displays the salt conditions as well as the voltage applied to either side (denoted by positive or negative signs) and how λ-DNA enters via EOF (anti-EPF) from the capture zone located at the outer walls of the nanopore. **b** Observation of CEs in asymmetric salt conditions when λ-DNA + 1 M KCl was added into the pipette and 4 M KCl was outside at −600 mV. To the right, EPF is used to repel λ-DNA away from the negatively applied voltage and exit the pore into the bath solution. Red inset: DNA exiting the pore produces CEs containing a tail before returning back to baseline. **c** Current traces of REs in asymmetric salt conditions when λ-DNA + 4 M KCl was added into the bath and the pore contained 1 M KCl. Located to the right is a schematic showing how DNA is electrophoretically attracted to translocate into the pore when +600 mV is applied. All buffers were prepared at pH 7.4. Blue inset: DNA entering the pore yields REs that look similar to square pulses. **d** Event decay to equilibrium for the case experiments shown in **b**. DNA is exiting the pore and thus should immediately leave the sensing zone of the pore as opposed to the reverse translocation direction.

sensing zone of the pore. Despite the DNA exiting the pore, there is a transient decay back to the baseline current level (Fig. 2b). Conversely, reverse translocations produced a square pulse rather than a pulse with a decaying tail (Fig. 2c). Figure 2b red inset and 2c blue inset provide examples of unconventional shapes seen under their respective conditions. By fitting the current to an exponential decay, the decay constant of forward translocations was found to be ~$1150 \pm 243$ s$^{-1}$, which corresponds to a 10% to 90% rise time of ~1.2 ms. This is substantially longer than the rise time associated with the 10 kHz lowpass filter used while recording the data (~33 μs)[50]. Although it is not clear as to what produces the observed waveform shapes, we speculate ion flux imbalance, its direction, and DNA translocation direction, to play a key role in the mechanism.

Conceptually, a pore can become ion-selective depending on its surface charge. If the pore is charge neutral, it would not exhibit any selectivity whereas if it is negatively charged, the pore would be cation-selective (Fig. 3a). Simulations with a negatively charged pore submerged in 10 mM KCl solution showed that although the pore's total ionic flux was not altered significantly by EOF (K$^+$ flux increased and Cl$^-$ flux decreased by the same amount), it does significantly impact the flux imbalance between cation and anion (see Supplementary Information for simulation details). The terms EOF- and EPF- pumping are used here to signify that ions are being moved by the insertion of electrical energy and energy is required to maintain the system in that state. Flux imbalance, defined here as |K$^+$ flux| minus |Cl$^-$ flux |, can be generated through externally applied conditions and parameters; for example, flux imbalance increases with both the pore diameter and the applied voltage (Supplementary Information Fig. 11). This finding will be important when discussing other monovalent salts wherein transitions between REs and CEs occur. Nevertheless, the finding that EOF can increase the counterion (K$^+$) flux imbalance is particularly noteworthy since (i) CEs were observed at high asymmetric salt conditions, which would also cause ionic flux imbalance and (ii) further supports the previous experimental observations of CEs occurring under asymmetric salt conditions as DNA translocates through the pore[21]. With a salt gradient, in addition to the electrical potential gradient, ions could move as result of the chemical potential gradient. Thus, for

in REs. Although directional dependence of DNA transport has been reported previously with nanopipettes[48], a change in the direction of the pulses has not been previously observed. The conductive pulse observations shown here showcase the shortcomings of theory used for nearly 2 decades, which presume excess charge introduced by DNA compensates for the ionic current blockade by DNA to eventually yield conductive pulses. Furthermore, with asymmetric salt conditions (1 M inside, 4 M outside), both the forward and reverse translocations produced unconventional event shapes. It is well-known that the translocation direction of a particle is reflected through its event shape with tapered pore geometries unlike their cylindrical counterparts[49]. Moreover, shapes analogous to that shown in Fig. 2b are typically observed for reverse translocations (i.e., when a molecule enters the pipette through the bath and travels along the confined tapered region). In other words, the geometry of the pore determines the electric field profile and by extension the

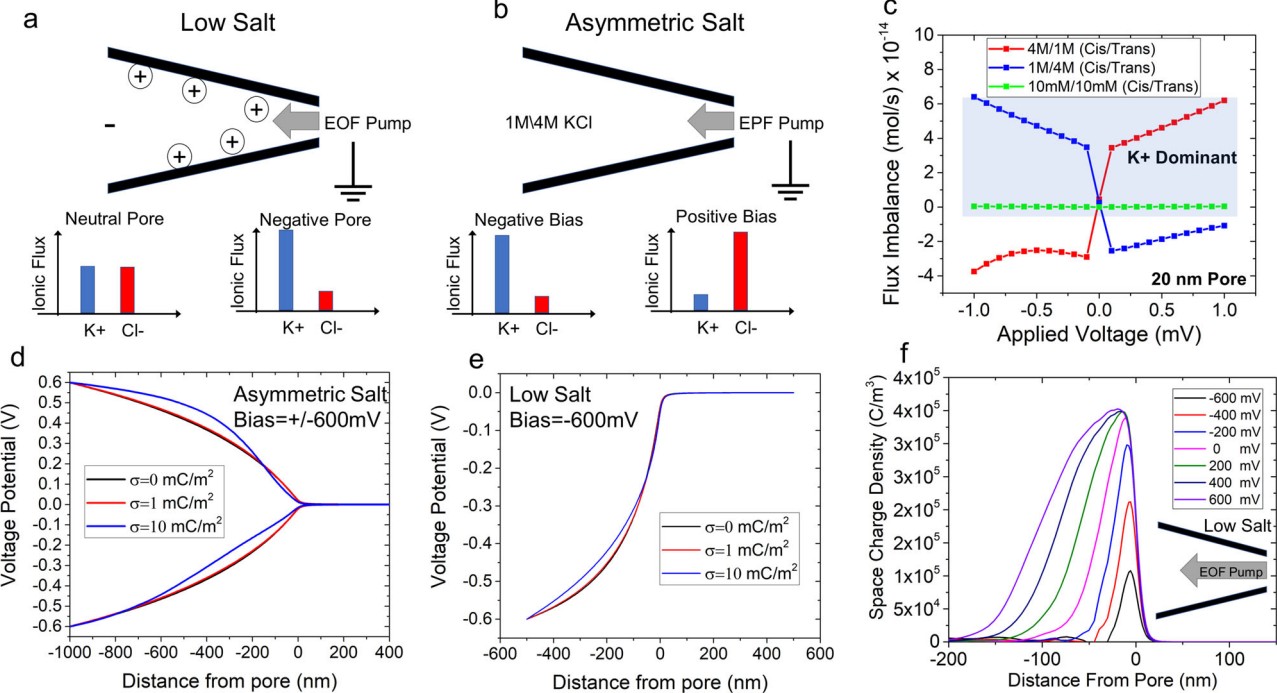

**Fig. 3 Conceptual and computational model of symmetric low-salt conditions and asymmetric salt conditions. a** A graphical representation of a negatively charged glass nanopipette under low-salt conditions. When a negative voltage is applied, EOF is directed into the pore The flux imbalances for a neutral pore and a negatively charged pore (our experiments) can be found at the bottom. Negatively charged pores enable a flux imbalance in favor of cations to occur when the pore has a negative potential. **b** An illustrative figure displaying EPF ion pumping for asymmetric salt conditions with 1 M KCl inside the pore and 4 M KCl outside. The graphs at the bottom represent a negative and positive voltage bias with the resulting flux imbalance. **c** Flux imbalance calculations for symmetric and asymmetric salt conditions (both conditions where 1 M (cis)/4 M (trans) and 4 M (cis)/1 M (trans) are shown). Asymmetric salt permits the toggling of the flux imbalance with either a change in voltage or concentration gradient formation. A blue shaded region is overlaid upon the region that is K$^+$ dominant. The symmetric low salt (10 mM/10 mM) curve is also provided in the Supplementary Information and shows that the pore is always cationic-selective. The potential distribution under **d** asymmetric salt conditions and **e** low-salt conditions for three surface charge densities (electric potential is plotted along the axis of symmetry). **f** Space charge density (C/m$^3$) for the voltage range of −600 mV to +600 mV (axial distance of zero corresponds to the tip of the nanopipette). The pore diameter for this simulation was 20 nm under low-salt conditions (10 mM KCl) and the schematic illustrates EOF pump directed towards the pore.

the asymmetric salt cases, assuming the same spatial voltage distribution, one ion will outweigh the flux of the oppositely charged ion, as shown in Fig. 3c. Note that 10 mM/10 mM conditions are also cation-selective at all voltages and is shown in Supplementary Information section 5. In case I, K$^+$ moves along both the electrical and chemical potential gradients opposing the DNA translocation direction whereas in case II, due to the positive bias, Cl$^-$ ions move along both the gradients cooperative with DNA translocation direction. This is also reflected through the translocation time ($\Delta t$) where case I produced events that were ~3× slower compared to case II ($\Delta t$ were 3.2 ms and 1.1 ms, respectively, for case I and case II). Taken together, these results imply a flux imbalance in favor of Cl$^-$ produces REs whereas CEs stem from a flux imbalance in favor of K$^+$. This is notably different than ion selectivity, which is typically a characteristic of the pore itself. Rather, flux imbalances can be generated through externally applied conditions and parameters. This computational finding further supports the previous experimental observations of CEs occurring under asymmetric salt conditions as DNA translocate the pore.

The impact of the flux imbalance seems to play a role in redistributing the voltage drop inside the nanopipette; in particular, the taper region where there is a confining negative surface. Using finite-element simulations, and varying the surface charge density incrementally, it is shown that higher surface

charges lead to two main effects. First, EOF flow velocity increases, and secondly, the excess charge inside the taper length of the pipette causes ion polarization effects. For example, as surface charge is increased, the electric potential drops significantly between −100 and −400 nm inside the nanopipette (Fig. 3d). Under asymmetric salt conditions, the impact is also voltage dependent since both EOF and EPF are voltage regulated; both producing a flux imbalance. Under positive voltages, the Cl- is rejected from the pore to the tapered region decreasing the voltage drop occurring inside the taper of the nanopipette. That is, as seen in Fig. 3, compared to a neutral pore surface, the tapered region become more conductive (i.e., less voltage drop occurs). On the other hand, if is a negative bias is applied inside the pore, K$^+$ ions are accepted to the pore interior causing the voltage drop inside the taper to increases compared to a neutral pore since the tapered region become more resistive (i.e., larger voltage drop compared to a neutral pore). Thus, the net positive or net negative charges stored inside the pipette changes the voltage distribution and therefore the sensing zone of the nanopipette sensor. The decrease in charge storage at low salt is observed in Fig. 3e wherein there is always a positive charge accumulation, but it is lessened or exacerbated by EOF. While EOF is the mechanism of charge transport, it is the flux imbalance that ultimately determines the degree of polarization. While charge density polarization effects are commonly taken

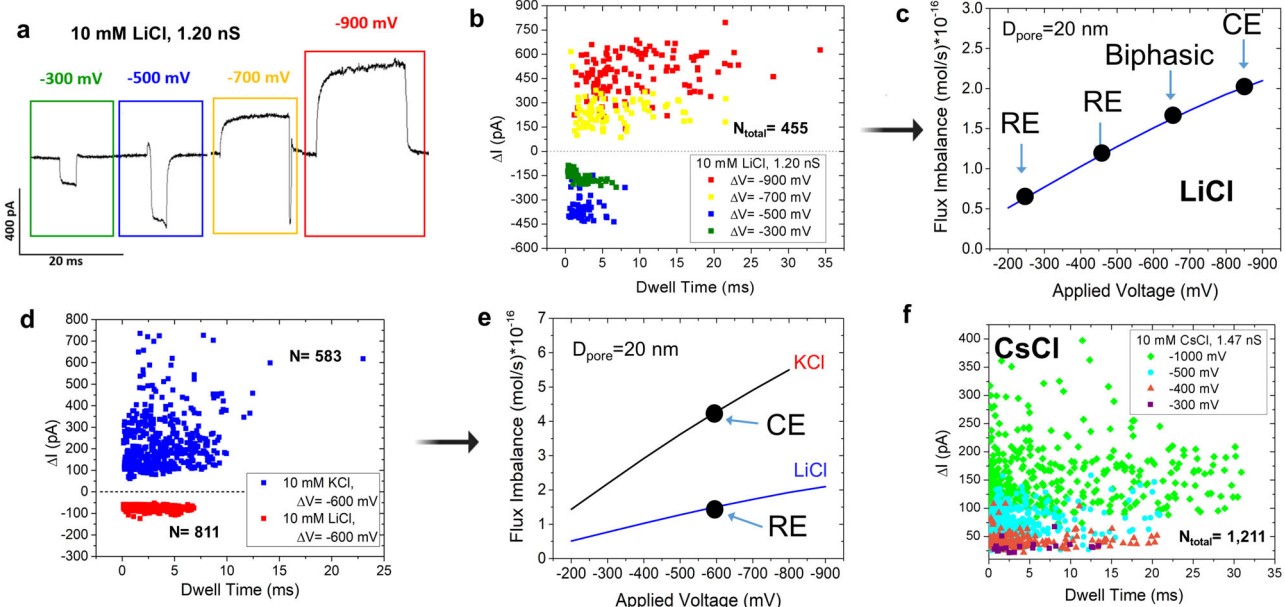

**Fig. 4 Event shape characteristics of λ-DNA via various monovalent salts. a** Representative waveforms observed in 10 mM LiCl from λ-DNA translocations in response to negative voltages. As the voltage increases in negativity, events transition from resistive to conductive. **b** Scatterplot showing current change and dwell time relationship with applied voltage for λ-DNA in 10 mM LiCl through a pore with a conductance of 1.2 nS at four different voltages: −900 mV in red, −700 mV in yellow, −500 mV in blue, and −300 mV in green. **c** Flux imbalance with (negative) applied voltage in 10 mM LiCl and its influence on the waveform generated through λ-DNA translocations. **d** An additional scatter plot corresponding to λ-DNA translocation in 10 mM KCl (CEs) and 10 mM LiCl (REs) in response to −600 mV. Both pores have a pore diameter estimated to be 33 ± 3 nm. The horizontal dashed line corresponds to the flux imbalance corresponding to the transition of REs to CEs. **e** Flux imbalance with (negative) applied voltage in 20 nm diameter pores in 10 mM KCl and LiCl. **f** Scatter plot of current change and dwell time corresponding to λ-DNA translocation in 10 mM CsCl in various applied voltages. All voltages produced CEs, similar to KCl.

into account on electrode-electrolyte interfaces, it seems rarely considered for nanoscale confinements until relatively recently[51,52].

**Alkali chloride dependence on event characteristics.** Now that a relationship between the ion flux imbalance and pulse direction is apparent, the question of whether the nature of the monovalent cation would have any effect on the transport properties was examined. For example, LiCl is known to shield the charge of DNA and slow it down compared to KCl since the former can bind more covalently to charged moieties compared to the latter[53]. Additionally, LiCl had a significantly higher streaming current compared to both KCl and CsCl (see Supplementary Information for more details). In this section, first, we draw comparisons between the translocation properties of λ-DNA in symmetric 10 mM KCl and 10 mM LiCl salts followed by 10 mM CsCl.

The nanopipette containing 10 mM LiCl was inserted inside a solution containing 10 mM LiCl and λ-DNA (buffered at pH~7.4) and current traces were recorded from −300 to −900 mV in 200 mV increments (Fig. 4a). As seen in Fig. 4a, a crossover from REs to CEs that is independent of salt concentration was observed. At voltages of −300 and −500 mV, λ-DNA translocations resulted in REs and at voltages of −700 and −900 mV, it resulted in CEs (also see Fig. 4c). Intrigued by this observation, we explored the pulse behavior at −600 mV where the event current shape assumed both a resistive and conductive region resembling a biphasic waveform (Supplementary Information Fig. 13). The biphasic nature of the events at the transitional voltages (−500, −600, and −700 mV) suggests that both resistive and conductive modulation mechanisms can conjointly act and perhaps act at different timescales in relation to the translocation event. For example, in the moments before or after the DNA enters the pore,

DNA would still exist within the EOF flow field of the pore, leading to current modulations on a potentially longer timescale.

Another comparison was done using two nanopipettes with inner diameters of 33 ± 3 nm. One nanopipette contained 10 mM KCl while the other contained 10 mM LiCl. Both were submerged in 10 mM LiCl with 500 pM λ-DNA (all buffered at pH ~7.4). Interestingly, at −600 mV, CEs were observed for the pore containing KCl whereas REs were observed for LiCl (Fig. 4e). At −600 mV, finite element simulations (for the 33 ± 3 nm nanopipette) predicted that the nanopipette is strongly cation-selective in KCl and weakly cation-selective in LiCl, which may be a possible explanation for the event types observed. If the transition to CE occurs at a flux imbalance of $2 \times 10^{-16}$ mol/s as shown in Fig. 4c, the same discriminating line appears to be valid for predicting KCl and LiCl current modulation (Fig. 4e). The stronger flux imbalance observed with KCl (under the same pore size, voltage, and salt concentration) led to CEs while LiCl produced REs (Fig. 4d, e). The critical value of the flux imbalance has no clear meaning at this time but is extracted from a combination of experimental and numerical approaches.

Intestingly, KCl had longer event durations at these low-salt conditions (3.1 ± 1.5 ms compared to 1.9 ± 0.7 ms in LiCl): a counterintuitive observation if DNA was electrophoretically driven since LiCl is known to slowdown DNA trasnlocation through charge shielding compared to KCl[53]. Since translocations in both KCl and LiCl are EOF driven, we suspect the effective charge shielding ability of LiCl allows EOF to transport the DNA with less opposing force. Other than the differences in Δt, as seen in Fig. 4d, the ΔI of REs observed for LiCl are much more tightly clustered together compared to ΔI of CEs observed with KCl (−70 ± 8 pA versus 200 ± 122 pA, respectively). The source of the variability of CEs observed in KCl is still not fully understood and requires further investigation. Once LiCl events trasition to

become CEs (Fig. 4b), current modulations become more scattered compared to REs. Additional information from λ-DNA translocating in 10 mM LiCl can be seen in Supplementary Information.

Recently, CsCl was shown to have an advantage over KCl in respect to sequencing using solid-state nanopores[11]. This publication used CsCl because it disrupts the hydrogen bonding between guanines, therefore denaturing the G-quadruplex into single-stranded structures. Although we are not working with ssDNA, we aimed to compare KCl event properties with another alkali metal chloride that holds promise in the nanopore community. Therefore, we performed experiments using nanopipettes filled with 10 mM CsCl inserted into 10 mM CsCl with λ-DNA (Fig. 4f). Similar to KCl, pulse direction in CsCl is expected to be voltage independent since $K^+$ and $Cs^+$ have nearly the same diffusion coefficient[54]. To confirm this, a pore with a conductance of 1.5 nS ($14 \pm 2$ nm diameter) was used under low ionic strength conditions and voltages of −300, −400, −500, and −1000 mV were applied. All voltages resulted in CEs. To further strengthen this observation, flux imbalance for CsCl was simulated (Supplementary Information), which revealed the pore to be cation-selective across the experimentally viable voltage range. Simulated results of KCl and CsCl were nearly identical due to nearly identical diffusion coefficients for $K^+$ and $Cs^+$ ($2.02 \times 10^{-5}$ and $2.00 \times 10^{-5}$ cm$^2$/s, respectively[54]).

**Protein conductive events at asymmetric salt conditions**. According to the experimental and numerical evidence, flux imbalance seems to play a role in producing CEs. Using the asymmetric salt conditions, we showed that a flux imbalance can be generated that favors potassium ions (i.e., case I). A reversed voltage polarity would therefore generate a flux imbalance that favors chloride ions (i.e., case II). We further wanted to investigate whether this would hold for protein structures since they notably have a heterogeneously charged surface. If analyte counterions played a role in CEs, we would expect cation and anion counterions would cancel out and there would be no observation of CEs. To study this, we chose to study the Cas9 mutant, Cas9d10a, because unbound it carries a net positive charge at pH ~7.4, and once bound to sgRNA, the complex becomes negatively charged[55]. For added specificity, amino acid sequence calculations were performed on the Cas9d10a complex alone and bound to sgRNA, providing net charges of both (Fig. 4a). The pH was not changed to be consistent with the previous set of experiments (e.g., same charge density on the pore and thus similar EOF). Furthermore, the same asymmetric salt conditions were employed, as before, where 1 M KCl was inside the nanopipette and 4 M KCl was outside the nanopipette. The Cas9d10a protein was added inside the nanopipette (in 1 M KCl) with and without sgRNA (resulting current traces are shown in Fig. 5b, c, respectively). The Cas9d10a-sgRNA complex was achieved by incubating Cas9d10a with sgRNA (equimolar amounts) for 1 h at room temperature. Voltages were applied to be consistent with the expected electrophoretic transport directions: positive bias for the Cas9d10a + sgRNA complex and a negative bias for the Cas9d10a protein. Like λ-DNA, Cas9d10a + sgRNA complex produced CEs (Fig. 5b). Under this condition, $K^+$ from the outside (4 M KCl) is driven into the pipette. However, upon reversing the voltage, the pore's flux imbalance was in favor of Cl- and thus Cas9d10a produced REs. In this condition, $Cl^-$ from the outside (4 M KCl) is driven into the pipette (referred to previously as EPF pumping of ions). The events at positive voltage could indeed be from either Cas9d10a + sgRNA complex or sgRNA alone since both are negatively charged. However, Cas9d10a binding of sgRNA is

typically fast with slowly reversible reaction kinetics[56]. The pulse direction is consistent with our previous observations where cation selectivity yielded CEs and anion selectivity yielded REs. It is also noteworthy to discuss the magnitude of the current enhancement between DNA and protein.

**Mechanistic insight into conductive events**. We have proposed a pore-centric model of CEs that is based on the dynamic distribution of ions inside of the nanopore. Volume exclusion is the typical mechanism of observing REs and we believe volume exclusion is still the main mechanism of CEs as well; both yield a transient ionic perturbation based on molecular occupancy of the pore. Since the voltage at the extreme ends of the fluidic reservoirs is clamped, charge build-up (i.e., potassium) tends to generate a voltage that, in turn, lowers the effective voltage for ion conduction at the pore. Inherent to a system with cation/anion flux imbalances is the concept of net neutrality, which is, by definition, violated by the conditions discussed here. Since electrostatics and ionic concentration profiles are coupled, voltage and ion flow are linked mechanistically. That is, especially with low electrolyte conditions, excess of either ion (cation or anion) could increase or decrease the voltage drop through the tapered region. The model developed for this study avoided the use of classical Nernst–Planck equations, which assume net neutrality. Instead, a Poisson–Nernst–Planck (PNP) model was developed, which permits ionic modulation of the electrostatic system. In the case of asymmetric salt conditions, the ion flux is also dependent on the chemical potential gradient where ions move from high salt to low salt generating a charge density polarization effect. In asymmetric salt, the pore can even be anion-selective, which is not possible under symmetric conditions. Depending on the voltage bias, the pore is either cation-selective or anion-selective, which changes the voltage drop in the tapered region and the pore. For the low-salt conditions, there the pore is always cation-selective since the quartz surface has a negative surface charge. The magnitude of the EOF is the critical factor that influences the current enhancement. For example, LiCl has less EOF (both in terms of average velocity and volumetric flow) in comparison to KCl and a transition to conductive events occurs at higher voltage (higher EOF). We speculate that a DNA-occupied pore transiently stops EOF (i.e., the effective pore size decrease during DNA occupation, which would result in diminished EOF) effectively lowering the charge stored inside the pore. Finite element methods demonstrate the accumulation of charge inside the glass pore (Fig. 3). The increase in stored charge with applied voltage is a characteristic trait of an ionic capacitor[57]. We believe that charge storage and dissipation dynamically impact the voltage at the pore therefore indirectly measures the occupancy of the molecule inside the pore.

An assumption used in the flux imbalance theory presented here is that occupancy of the DNA or protein leads to less polarization through disturbing the equilibrium conditions of the open-pore. For nanopore conditions in which a flux imbalance is created by convective flow, it is easy to see how a translocating entity can block fluid flow. For asymmetric salt conditions, the role of osmotic flow and its role in generating a flux imbalance is an important area that needs exploration. Nevertheless, even for conditions with no fluid flow, the mere reduction of ionic flow (equal reduction of $K^+$ flux and $Cl^-$ flux) may reduce the polarization of the nanopore. Based on the decay rate of events (Fig. 2), it seems that polarization is in dynamic equilibrium and, furthermore, associated with a time constant. A second point to consider is the role of the nanopore geometry. Based on the asymmetric salt conditions that were studied, a $K^+$ flux imbalance into the nanopipette seems to yield the greatest polarization effect

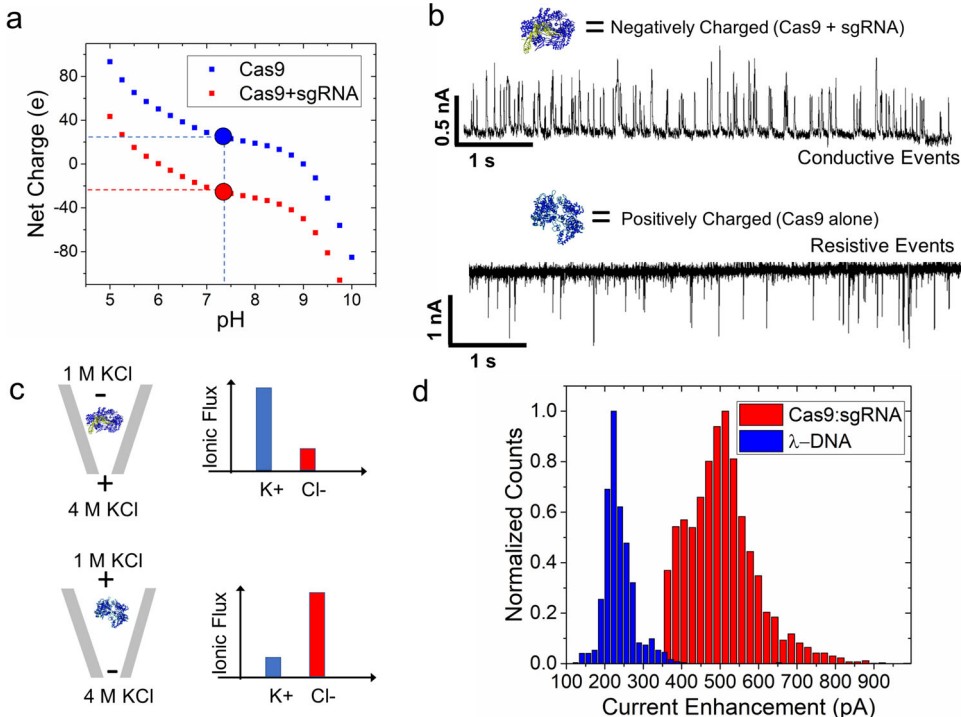

**Fig. 5 Event characteristics for Cas9d10a and the Cas9d10a + sgRNA complex under asymmetric salt conditions (1 M inside pore/4 M KCl outside pore). a** Graph of net charge based on calculations of amino acid composition of Cas9d10a alone and Cas9d10a-sgRNA at various pH values. The black dotted line represents the pH value of our working conditions (7.4), and the red and blue circles are the corresponding charges of Cas9d10a + sgRNA and Cas9d10a alone, respectively. **b** Current trace of Cas9d10a with and without sgRNA in asymmetric salt conditions (1 M KCl inside and 4 M KCl outside). Events were resistive when Cas9d10a was inside the nanopipette, and a positive voltage is applied inside the nanopipette. Cas9d10a was pre-incubated before diluting in 1 M KCl where the Cas9d10a and sgRNA were in an equimolar ratio (1:1). Events were conductive when Cas9d10a + sgRNA was inside the nanopipette and a negative voltage was applied inside the nanopipette. **c** Schematic of set-up (left) and fluxes of K⁺ and Cl⁻ under both conditions. Top condition includes the negatively charged complex of Cas9d10a and sgRNA. The bottom condition contains the positively charged Cas9d10a molecule alone. Both have 1 M KCl + analyte within the nanopipette and 4 M KCl within the electrolyte bath solution. **d** Current enhancement observed for both DNA and protein using similar sized pores (36 nS for Cas9:sgRNA and 33 nS for λ-DNA) and the same voltage bias of −500 mV. An asymmetric salt condition was used on both experiments (1 M/4 M KCl) and voltage was applied inside the nanopipette; driving negative DNA and protein–RNA complexes out of the nanopipette.

that led to a greater current enhancement for DNA translocation. K⁺ flux out of the nanopipette did not achieve the same level current enhancement Upon DNA entering the pore. The rationale that positive charge can be stored in the negative taper length of the nanopipette is used to explain the high current enhancements at this condition: 1 M + DNA inside the pipette, 4 M outside.

Ionic-generated potentials are typically named according to the principle in which they are generated. For example, diffusion potentials, streaming potentials, and exclusion potentials[58]. Nevertheless, charge separation is a commonality of these potentials as well as our capacitor model, which ultimately could generate voltage and current transients. Data thus far support the hypothesis that a flux imbalance plays an important role in the generation of CEs and the evidence here demonstrates the importance of the pore's charged surface, voltage-bias, and associated electro-hydrodynamics in generating CEs. In this study, we described multiple electro-hydrodynamic effects that influence EOF-driven DNA translocations under low ionic strength conditions. We have found that EOF can be used in various alkali chlorides. Confirmation that EOF capture volume resides along the sides of the tip aperture and directs flow inward has been shown. The resulting current enhancement or reduction dependence on pore size can be explained by a pore's flux imbalance. Secondly, we discovered a pulse crossover point from CEs to REs, independent of salt concentration and specific to

LiCl, by scanning the applied voltage from −300 to −900 mV. We show that changing the electrolyte influences the event shape, SNR values, and event frequency. The pulse nature was also explored for proteins with Cas9 mutant, Cas9d10a, in both free form and bound to sgRNA wherein CEs were observed for the Cas9d10a- sgRNA complex and REs were observed for the free Cas9d10a protein. The pulse direction results were in good agreement with the flux imbalance theory proposed for DNA. Evaluating polarization effects and its role in producing CEs will provide a framework for understanding experimental results at these low salt and asymmetric salt conditions. Therefore, we propose an additional possible theory for conductive events based on charge density polarization where accumulation of positive charge (for a negatively charged pore), via a flux imbalance, appears to effectively lower the voltage bias at the pore during open pore conditions and enhances the current when the equilibrium conditions are altered.

## Methods

**Nanopore preparation**. Nanopore fabrication began with quartz capillaries (Sutter Instrument Co.) of 7.5 cm in length, 1.00 mm in outer diameter, and 0.70 mm in inner diameter. Capillaries were plasma cleaned for 5 min before laser-assisted machine pulling to remove any surface contaminations. Afterwards, quartz capillaries were placed within the P-2000 laser puller (Sutter Instrument Co.) and a one-line protocol was used: (1) HEAT: 630; FIL: 4; VEL: 61; DEL: 145; PULL: between 135 and 195. This resulted in two identical, conical nanopores. The heat duration was ~4.5 s.

Electrodes were constructed using silver wires dipped in bleach for 30 min followed by thorough rinsing with water to remove any residual bleach. Freshly pulled nanopipettes were then backfilled with either 10 mM KCl (Sigma Adlrich), LiCl (Sigma Adlrich), or CsCl (Alfa Aesar) buffered at pH~7.4 using the Tris-EDTA buffer (Fisher BioReagents). The conductivities of each alkali chloride were recorded using an Accumet AB200 pH/Conductivity Benchtop Meter (Fisher Scientific). The results were as follows: 10 mM KCl = 0.26 S/m, 10 mM LiCl = 0.23 S/m, and 10 mM CsCl = 0.26 S/m at room temperature. An optical microscope was used to inspect the nanopipettes at this stage for any irregularities. Once the nanopipettes had been inspected, electrodes were connected to the head stage of the Axopatch 200B (Molecular Devices).

**Data acquisition**. The Axopatch 200B patch-clamp amplifier was used in voltage-clamp mode to measure the ionic current changes. The gain was optimized before each experiment and the signal was filtered with the inbuilt low-pass Bessel filter at 10 kHz and digitized using Digidata 1550B (Molecular Devices). The data was acquired at a frequency of 250 kHz. Data analysis for DNA translocations and folding were performed using a custom MATLAB code.

**Finite elements methods**. COMSOL Multiphysics was used for modeling nano-pipette geometries that were based on scanning electron microscope (SEM) and transmission electron microscope (TEM) images acquired from the same pipette pulling protocols that were used in sensing experiments. A two-dimensional (2D) axisymmetric model was employed to reduce the computational resources required. Once the geometries were created in COMSOL, the physics that were utilized included Poisson–Nernst–Planck–Stokes equations: laminar flow, transport of diluted species, and electrostatics. The electrostatics boundary condition for the glass was set at a surface charge density of $-2 \times 10^{-2}$ C/m$^2$. To model electroosmotic flow, a volume force on the fluid was set to the space charge density of the ions in solution multiplied by the electric field vectors (r and z vectors). An inbuilt EOF boundary condition was also tested and yielded similar results. Diffusion coefficients and mobility values were obtained from Lee et al.[54] All models were tested with different solvers, solving conditions, and reservoir sizes to ensure the accuracy of results. The Stokes flow boundary conditions were no-slip, and the inlet and outlet were kept at the same 1 atm of pressure, which is consistent with experiments. The z-component of the flux was extracted for each model from a 2D line that spans the width of the pore. The flux was then integrated across this 2D line to obtain the flux in moles/s.

## Data availability

Data will be available from the authors upon request and approval.

## Code availability

Code will be made available upon request and approval.

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

## Acknowledgements

Software suits were provided by the University of California, Riverside.

## Author contributions

K.J.F. formulated the idea to explore low ionic strength solutions using DNA and nanopores. L.S.L. performed the experiments, devised analysis protocols, and carried out data analysis under the guidance of K.J.F. M.N. performed preliminary experiments. N.F. simulated experiments using COMSOL. Y.M.N.D.Y.B performed the transferrin experiments. L.S.L, Y.M.N.D.Y.B, and K.J.F. wrote the manuscript.

## Competing interests

The authors declare no competing interests.
