## [Peer Review File · Nature Communications]

Peer review comments, first round review –

Reviewer #1 (Remarks to the Author):

In this manuscript, Lastra and co-workers describe a comprehensive study of ionic current blockades produced by the translocation of charged biomolecules through nanocapillaries. In comparison to many previous studies that investigated similar translocation processes, this study stands out by reporting a meticulous examination of the ionic current change (blockade or enhancement) as a function of the pore geometry, electrolyte concentration and composition, but also asymmetry, and voltage polarity and magnitude. Furthermore, the study examines blockades produced by both dsDNA and charged protein-RNA complex. The authors rationalize their finding using continuum model calculations and the “flux imbalance” considerations.

The manuscript leaves a mixed impression.

PROS: The study is systematic, examines a variety of systems and is, in a way, focused on just one thing: showing that flux imbalance plays an essential role in determining the magnitude of an ionic current change when a charged molecule passes through a nanopore. The manuscript reports plenty of new and exciting experimental data, in particular those described in Figures 2, 4 and 5.

CONS:

1: The manuscript is difficult to read. The problem is not spelling or grammar, but imprecise writing, use of hyperboles, and missing logic. The figures, in particular figures 1 and 3, are not organized in a way that makes the results obvious. The captions are also too short for Nature Communications and the figures cannot be understood on their own. Finally, the writing becomes tedious and speculative closer toward the end of the results section.

2: The authors start the manuscript with setting an ambitious goal of determining the origin of the current blockades (the title is even more grandiose than that). But, at the end of the day, have the authors accomplished that? Armed with the new insights into the physics of the process offered by the new experimental data, can the authors predict or at least quantitatively reproduce the experimental blockade data? Unfortunately, this does not appear to be the case. While one can criticize the simplistic “current enhancement model” (hypothesis 1 in the introduction) or be skeptical about the effect of local mobility reduction (hypothesis 2 in the introduction, which is an improvement of hypothesis 1), those two models provide a straightforward recipe to computing a current change knowing the configuration of the molecule in the nanopore and bulk ion concentration. The authors should provide a similar computational model that transforms the “flux imbalance arguments” into quantitative predictions and compare the results of such model to (at least) their own experimental data. Ideally, the authors should show the failure of the traditional model and the superiority of the model that accounts for “flux imbalance”.

If the authors are not able to develop a mathematical model, they should change the emphasis of the manuscript from “determining the origin” to “illustrating the effect of”. The manuscript should be reviewed again with the new scope in mind.

The authors may find the following detailed comments on the manuscript useful when preparing the revisions.

Abstract has too many abbreviations to be readable as a standalone summary of the work.

Line 49: Does CE stand for “current enhancing” or conductive event”? Probably the former, but writing is confusing.

Line 50-68. I think the authors have misinterpreted the literature. There are no two separate hypotheses about two separate mechanisms, i.e., an increase of ion concentration or effects of

friction. Both are correct and occur at the same time, as described in Ref 3. That seems to be the consensus in the field.

Lines 69-84. The logic is hard to follow. Should the reader associate anion and cation-selective conditions with REs and CEs? That's not obvious. Also, please also define what "flux imbalance" and "imbalance of ionic fluxes" (line 99) mean and how it is different from "ions selectivity".

Line 87: The EOF-driven transport of DNA was first predicted in J. Phys.: Condens. Matter 22 (2010) 454123, well ahead of the cited references. Please correct this omission.

Figure 1 needs a schematic defining the polarity of the bias, the direction of the EPF and EOF forces, the direction of the ionic current and EOF with respect to the pore geometry.

Figure 1b: please specify which part of the dependence was used for the linear fit and Eq. 1

Figure 1c is incomprehensible. Please draw a schematic of the continuum model defining what "the distance to pore" is. Is it along the capillary or radial? If the former, what was the pore size, cone angle, how do the values depend on the two? One can add one set of extra traces without complicating the plot. Also, the fluid velocity is not uniform and the DNA occupies regions of different flow magnitude. What happens then to the molecule, does it stretch, breaks? Different parts of the molecule cannot move with different velocities for very long.

Line 156: Define what "operational conditions adopted in this work" are. The prior description lists a range of mutually exclusive situations.

Figure 1d. What is "pump velocity"? Is the pump moving somewhere? What is the meaning of the white lines? What is actually shown in the inset? The caption says fluid velocity, but it appears to be an image of something labeled.

Figure 1 is missing several examples of current blockades (or enhancements) associated with the EOF-driven translocation events.

Consider flipping the orientation of Figure 1d to be consistent with 1e.

The data shown in Figure 1e are really interesting! Still, this reviewer finds it very surprising that the capture volume extends to millimeters. Are there any theoretical calculations that would support such a macroscopic range, which is three orders of magnitude larger than that for EP capture?

Have the authors ruled out that what is seen in Figure 1e relates to capillary pressure or partially filled pipette? Why not submerge the entire capillary in the fluid?

Line 180: "...nanopore was suspended from a micrometer" What does this mean?

SI does not seem to be organized to follow the flow of the main text.

It is not clear why the following is assumed to be true, please elaborate or better provide some simple mathematical arguments: "If counterions become sheared off the negatively charged DNA or overlapped with the counterions from the pore wall to enhance ion concentrations inside the pore, CE amplitude should increase with decreasing pore size".

Trace in Figure 2a is the same as in S7. Reusing the trace leaves an impression that that was the only trace collected.

Figure 2 "Observations on the nature of events ..." sounds very esoteric. Is it possible to have a more straightforward title? Just describe the experiment. In general, the word "Observations" cannot describe what is shown, it is a process.

Back in 2009, all-atom MD simulations have shown that current enhancement can be seen in high

molarity experiments, see Figures 11 and 12 of Biophysical Journal 96:593-608 (2009). Pore geometry and non-homogeneous ion distributions can affect the blockade level in a very non-trivial way. The authors should discuss their finding in the context of that work.

Line 206: What exactly is "directional dependence of DNA transport"? That DNA can move in both directions when one switches polarity of the field? Probably not what the authors meant, but what did they mean?

Line 207 "a change in the conductive nature of the pulses" ... confusing, better use simple words like a transition from RE to CE upon reversal of bias polarity or something similar.

Line 212 "produced anti-conventional event shapes" What is opposite of conventional? Unconventional? In what way? Perhaps the authors should show both "conventional" and "anti-conventional" shapes to give a reader a chance of grasping the meaning. Why is that important?

Figure 3 and the rest of the manuscript. Unfortunately, the authors never defined what "flux imbalance" means, which makes the rest of the manuscript difficult to understand. Is it the same as ion selectivity, which is the difference in the magnitude of the currents carried by different ionic species? If so, it is not at all surprising that moving fluid will change the contribution of ionic species to the total current. Or does it refer to some transient, non-equilibrium phenomena leading to ion cloud polarization because of the finite volume of the system? How does one transform the latter into a quantitative prediction about the magnitude of the ionic current blockade? If that is the case, the finite size of the volume must be explicitly taken into account.

Reviewer #2 (Remarks to the Author):

The manuscript by Lastra et al. reports on new, detailed studies of DNA and protein translocation through nanochannels. In particular, they investigate the role of electroosmotic flow (EOF) and electrophoretic (EP) transport on the direction of transport, both experimentally and via simulations. While this has been studied to some level of detail for proteins, the authors correctly point out that for DNA this is severely understudied. Indeed, one key finding of the study is that also in the case of DNA the direction of transport can be determined by EOF, rather than EP. Interestingly, the authors then also go on to propose a new model for how the event characteristics emerge, highlighting the importance of polarisation effects during temporary channel blockage. Concentration polarisation is of course a common phenomenon and routinely considered in membrane science, but it is indeed new and very interesting in the present context. This is because our physical understanding of the origin of resistive/conductive pulse sensing is closely linked with the interpretation of the results and, ultimately, also to the capabilities of the sensor. For example, it provides a new perspective on the interpretation of the relation between translocation time and DNA length, the spatiotemporal resolution of the sensor and how sub-structure in DNA are detected (presumably when polarisation effects are minimised?). It is becoming clear that the actual current-time events are a convolution of multiple effects, depending on the conditions, which may include volume exclusion, ion condensation and, according to the authors, also concentration polarisation. Accordingly, the authors present a range of arguments in support of the model, which warrants further investigation.

Formally, the work is of a high standard throughout. The text is generally written well, even though there are some typographic errors and misprints (p. 4, l. 74 "stokes" should be capitalised; p. 7, l. 137 "oriface" or l. 141 "aparature"; p. 17, l. 297 "Res" should be "REs", several on p. 18 and so on). Some more careful proof-reading is advised before submitting a revised version of the manuscript. On several occasions, references are also missing, for example on p. 6, l. 130 (where does the value for the surface charge come from?) or on p. 7, l. 155 (the value for the electrophoretic mobility of DNA). The terminology around which electrode is anode and cathode needs to be corrected (p. 7, ll. 143), since also the grounded electrode will act as the respective other (depending on the bias applied to the other electrode).

Overall, I think this work should be of great interest to the community and, subject to minor

revisions, can be considered for publication in the journal. It clearly gives new impulses to the field and highlights the potentially more general importance of "dynamic" effects in the context of nanopore sensing.

Reviewer #3 (Remarks to the Author):

The authors report a very exciting study on glass capillary nanopores, a very interesting technique to probe single molecules (e.g. DNA) and to study Physics at the Nanometer Scale. Compared to many other (cited) works, they apply a low salt concentration in the buffer solution which enhances electrostatic effects. For the first time (to my knowledge) they systematically study transport of DNA molecules and fluid caused by a salt concentration gradient. The experimental work is accompanied by simulations based on the electrokinetic equations, a set of partial differential equations describing the coupled interaction and motion of ions and the surrounding fluid.

The applied methodology is sound and in line the state of the art (to my knowledge). The rich complexity of the physics leads to an interesting spectrum of observed conducting and resistive events (CE/RE) where the presence of a DNA molecule in the pore modulates the current towards higher or lower values. The works clearly show that the methodology is well suited to gain understanding of the physics at that length scale. The authors argue that the flux imbalance, i.e. asymmetric contribution of both ion species to the conductivity due to electrostatic exclusion of one ion species is a key factor to explain the observations. I am very supportive of the author's idea to disentangle the different physical effects, and to find intuitive explanations for this (to my knowledge) still unsolved puzzle. But the author's conclusion that flux imbalance explains most the observations is too short. A combination of flux imbalance with other effects (e.g. electroosmotic flow) is required to cause the effect. Furthermore, it is important to stress, that many of the observed effects are effects non-linear in the applied voltage (e.g. the inversion of the event in Fig 4a), and this should be discussed in more detail.

From my perspective the (difficult) theoretical analysis is a weakness of the manuscript. From my point of view, this is reflected in the fact that a few aspects and concepts are missing. First, this includes the discussion of linear vs. nonlinear response, and the respective perturbation approaches in the literature. Furthermore, this includes the following points.

The authors introduce the concepts of electroosmosis and electrophoresis well, but do not mention the main quantity which is typically used to quantify the (effective) magnitude of EOF and electrophoretic mobility, the zeta potential. It unites the concepts of electrophoresis and electroosmosis, which is very useful as it is governed by identical physics. In their comparison, the authors compare the electrophoretic mobility of DNA which includes an effective surface charge and a glass charge density where it is not clear if this how this quantity was measured.

The other fundamental quantity that is not introduced is the Debye length which describes the range of interactions between a charged surface and the surrounding solution. The experiments are very interesting as, other than many works, the low salt regime leads to a situation where the Debye length is comparable to the pore diameter leading to conditions where direct electrostatic and electrokinetic interactions of the DNA with the pore become relevant.

Furthermore, the authors perform very interesting experiments with asymmetric salt conditions, which gives rise to osmosis, where diffusion ions draw fluid along through the pore. The authors do not consider this effect even though it appears to me to be not very difficult to include into the simulation as only boundary conditions need to be altered. Then, the resulting fluid flow magnitude could be compared to the electroosmotic result indicating how this affects the fluid flow profiles.

The observed tails in conductive events are very exciting as they indicate that "something nonlinear" is happening. As the tails only appear when the osmosis opposes the electrophoretic DNA transport, I would speculate that the DNA does not leave the pore orifice region as fast as when osmosis is not present. The authors speculate that it is a transient effect of the electrokinetics. An upper bound for the time scale of relaxation from the diffusion constant of the ions ($\sim 10^{-9}$

m^2/s) and the relevant length scale of the distortions (which is difficult to pick carefully). Assuming a relevant length scale of 100 nanometer, I obtain a relaxation time of 10 microseconds. Transient effects of the fluid are governed by the kinematic viscosity ($\sim 10^{-6}$), which leads to an even faster time scale. Therefore I have doubts regarding this explanation.

In summary, the authors have performed very exciting work towards understanding the physical mechanisms. But I recommend reiterating the theoretical discussion. Maybe, from simulation find analogies, systematically study the transition from linear to non-linear response, etc this is possible - although I personally experienced it being difficult. I have named a few concepts I believe should appear in the discussion.

And finally two small remarks.

The authors deduce the pore diameter from conductivity measurements. They state they infer the conductivity from the slope of the I-V curve. Obviously, the curve is nonlinear, and this makes me wonder how they have taken the slope. I think this is worth commenting on. This includes the error bars on pore diameters (l. 121). I believe that the nonlinearity of the curve violates the assumptions of the diameter formula (1) the curve still should create the right qualitative picture. As the authors confirm the diameter measurements with TEM, the method is valid.

l. 220 "tale" -> "tail"

REVIEWER COMMENTS

Reviewer #1 (Remarks to the Author):

In this manuscript, Lastra and co-workers describe a comprehensive study of ionic current blockades produced by the translocation of charged biomolecules through nanocapillaries. In comparison to many previous studies that investigated similar translocation processes, this study stands out by reporting a meticulous examination of the ionic current change (blockade or enhancement) as a function of the pore geometry, electrolyte concentration and composition, but also asymmetry, and voltage polarity and magnitude. Furthermore, the study examines blockades produced by both dsDNA and charged protein-RNA complex. The authors rationalize their finding using continuum model calculations and the “flux imbalance” considerations.

PROS: The study is systematic, examines a variety of systems and is, in a way, focused on just one thing: showing that flux imbalance plays an essential role in determining the magnitude of an ionic current change when a charged molecule passes through a nanopore. The manuscript reports plenty of new and exciting experimental data, in particular those described in Figures 2, 4 and 5.

CONS:

1: The manuscript is difficult to read. The problem is not spelling or grammar, but imprecise writing, use of hyperboles, and missing logic. The figures, in particular figures 1 and 3, are not organized in a way that makes the results obvious. The captions are also too short for Nature Communications and the figures cannot be understood on their own. Finally, the writing becomes tedious and speculative closer toward the end of the results section.

Response to Reviewer:

Thank you for taking the time to review our manuscript. The overall cohesiveness of the manuscript has been improved (further editing in regards to removing the imprecise writing, hyperboles, and logical errors) and all captions have been expanded upon so they can be understood alone.

In regards to the figures, modifications have been made to both. Specifically in Figure 1, the orientation of 1d has been flipped to match the pore orientation in 1e. A different schematic displaying the direction of electroosmotic flow (EOF) and electrophoretic force (EPF) under negative applied voltages (yellow-shaded region) has replaced the old schematic in 1b. Additionally, more information has been provided in the caption of this figure to help make the results more obvious. For Figure 3, the schematics in a and b have been color coded to match what c has displayed to aid readers in understanding the message we are conveying. Further clarification and explanation has also been added to Figure 3's caption.

Figure 1

Figure 1: Experimental set-up and characterization of quartz nanopores. (a) TEM of quartz nanopore; scale bar, 50 nm. (b) I-V curves pertaining to four differently sized nanopipette

orifices. For pore size estimations, the linear portion at the negative voltages was used (yellow shaded region). The schematic within the I-V curves shows the directionality of EOF and EPF at negative voltages. (c) EOF, EPF drift, and the resulting net velocities of λ -DNA along the pore's axis of symmetry ($\mu=3.2 \times 10^4$ cm/Vs). Distance from the pore is radial from the axis of symmetry. (d) Simulations of fluid flow velocities under low ionic strength conditions. White lines indicate fluid flow lines for a 20 nm pore at -600 mV voltage bias. Inset: YOYO-labelled DNA sample with an applied voltage of -700 mV to visualize the capture zone. The gray line at the center indicates the pore's axis of symmetry, which aids in deciphering the distance from the pore simulation results provided in (c). (e) Event frequency with depth of the pipette inside the bath solution. Nanopore depth is synonymous with how deep the nanopore tip was submerged into the analyte-containing bath solution. (f) Linear DNA events from 17 pores were investigated for pore size dependence on current amplitude. We see that the enhancements fluctuate between 60 and 140 pA with no discernable trend.

Figure 3

Figure 3: Conceptual and computational model of symmetric low salt conditions and asymmetric salt conditions. (a) A graphical representation of a negatively charged glass nanopipette under low salt conditions. When a negative voltage is applied, EOF is directed into the pore (as shown by the blue arrow). The flux imbalances for a neutral pore and a negatively charged pore (our experiments) can be found at the bottom. Negatively charged pores enable a flux imbalance in favor of cations to occur when the pore has a negative potential. (b) An illustrative figure displaying EPF ion pumping for asymmetric salt conditions with 1 M KCl inside the pore and 4 M KCl outside. The graphs at the bottom represent a negative and positive voltage bias with the

resulting flux imbalance. (c) Flux imbalance calculations for symmetric and asymmetric salt conditions (both conditions where 1 M (cis)/4 M (trans) and 4 M (cis)/1 M (trans) are shown). As seen in the figure, it is possible to toggle the imbalance of fluxes with either a change in voltage or concentration gradient formation. The potential distribution under (d) asymmetric salt conditions and (e) low salt conditions for three surface charge densities (electric potential is plotted along the axis of symmetry). (f) Space charge density (C/m³) for the voltage range of -600mV to +600 mV (axial distance of zero corresponds to the tip of the nanopipette). The pore diameter for this simulation was 20 nm under low salt conditions (10 mM KCl). For simplicity, boxes outlined in light green pertain to low salt information while outlines in blue represent asymmetric salt (1 M (cis)/ 4 M KCl (trans)).

2: The authors start the manuscript with setting an ambitious goal of determining the origin of the current blockades (the title is even more grandiose than that). But, at the end of the day, have the authors accomplished that? Armed with the new insights into the physics of the process offered by the new experimental data, can the authors predict or at least quantitatively reproduce the experimental blockade data? Unfortunately, this does not appear to be the case. While one can criticize the simplistic “current enhancement model” (hypothesis 1 in the introduction) or be skeptical about the effect of local mobility reduction (hypothesis 2 in the introduction, which is an improvement of hypothesis 1), those two models provide a straightforward recipe to computing a current change knowing the configuration of the molecule in the nanopore and bulk ion concentration. The authors should provide a similar computational model that transforms the “flux imbalance arguments” into quantitative predictions and compare the results of such model to (at least) their own experimental data. Ideally, the authors should show the failure of the traditions model and the superiority of the model that accounts for “flux imbalance”.

Response to Reviewer:

The strongest argument that we have provided in the manuscript for the failure of these models is that they do not predict current enhancements for any other molecule aside from DNA at low salt KCl. For example, low salt conditions with LiCl (at the same voltage and pore), did not show conductive events. We also do not believe that current models can explain how DNA and protein have conductive events at high asymmetric salt conditions. Protein conductive events perhaps are the most troublesome to the current school of thought since they have a heterogeneous charge on the surface of the molecule.

Therefore, the current theory of the introduction of counterions via a homogeneously charged DNA molecule seems to only work for exactly those conditions (DNA in low salt KCl) which left us doubting this as the actual true model. We find it unlikely that conductive events have a different mechanism for DNA at low salt, compared to every other condition that produces conductive events. In terms of observing conductive events for other molecules at low salt, PEG molecules also generate conductive events and we expect PEG (a neutral polymer) to have no

counter ions. We also observe conductive events with protein (both at low and asymmetric salt conditions). We have since added a current trace of protein at low salt in the SI which also shows conductive events. The magnitude of the current enhancement (for asymmetric salt) for the protein Cas9 was also quite larger than DNA which points to some other mechanism besides counterions. For example, homogeneously charged DNA should definitely introduce more counterions than a heterogeneously charged molecule since the counterions are of mixed valency (+1, -1).

The proposed model on the other hand points to a mechanism that more strongly relies on the nature of the pore, rather than the molecule itself. The flux imbalance models we use do not include the molecule at all. We did not include the molecule since simulations that place a stationary molecule into the pore are unrealistic in many ways. The model still is able to predict both low salt and asymmetric salt conductive event behavior and therefore, it is because of this evidence that we propose, indeed, the superiority of the flux imbalance model.

Manuscript Changes: Clarification of the proposed model is elaborated on in the following sections:

“How a flux imbalance yields CEs is yet to be addressed to date despite being a commonly observed phenomenon. Since the voltage at the extreme ends of the fluidic reservoirs is clamped, charge build-up (i.e., potassium) tends to generate a voltage that, in turn, lowers the effective voltage for ion conduction at the pore. That is, especially with low electrolyte conditions, the ion selectivity of the pore could either increase or decrease the voltage drop through the tapered region by accepting (i.e., K^+) or rejecting (Cl^-) ions through EOF. In the case of asymmetric salt conditions, the ion flux is also dependent on the chemical potential gradient where ions move from high salt to low salt generating a charge density polarization effect. In asymmetric salt, the pore can even be anion selective which is not possible under symmetric conditions. Depending on the voltage bias, the pore is either cation selective or anion selective, which changes the voltage drop in the tapered region and the pore. For the low salt conditions, there the pore is always cation selective since the quartz surface has a negative surface charge. The magnitude of the EOF pumping is the critical factor which influences the current enhancement. For example LiCl has lesser EOF in comparison to KCl and a transition to conductive events occurs at higher voltage (higher EOF). We speculate that a DNA-occupied pore transiently stops EOF pumping (i.e., the effective pore size decrease during DNA occupation which would result in diminished EOF pumping) effectively lowering the charge stored inside the pore. Finite element methods demonstrate the accumulation of charge inside the glass pore (Figure 6b). The increase in stored charge with applied voltage is a characteristic trait of an ionic capacitor⁵⁵. We believe that charge storage and dissipation dynamically impact the voltage at the pore therefore indirectly measures the occupancy of the molecule inside the pore.

An assumption used in the flux imbalance theory for conductive events is that occupancy of the DNA or protein leads to less polarization through disturbing the equilibrium conditions of the open-pore. For nanopore conditions in which a flux imbalance is created by convective flow, it is easy to see how a translocating entity can block fluid flow (as well as ionic flow). For asymmetric salt conditions, the role of osmotic flow and its role in generating a flux imbalance is an important area that needs exploration. Nevertheless, even for conditions with no fluid flow, the mere reduction of ionic flow (equal reduction of K^+ flux and Cl^- flux) may reduce the

polarization of the nanopore. Based on the decay rate of events (Figure 2), it seems that polarization is in dynamic equilibrium and, furthermore, associated with a time constant. A second point to consider is the role of the nanopore geometry. Based on the asymmetric salt conditions that were studied, a K⁺ flux imbalance into the nanopipette seems to yield the greatest polarization effect which led to a greater current enhancement for DNA translocation. K⁺ flux out of the nanopipette did not achieve the same level current enhancement Upon DNA entering the pore. The rationale that positive charge can be stored in the negative taper length of the nanopipette is used to explain the high current enhancements at this condition: 1M + DNA inside the pipette, 4M outside.”

If the authors are not able to develop a mathematical model, they should change the emphasis of the manuscript from "determining the origin" to "illustrating the effect of". The manuscript should be reviewed again with the new scope in mind.

Response to Reviewer:

Most of our response to this comment can be addressed above. In summary of the above, we propose a mathematical model that explains conductive events based on the conditions of the pore itself and the buffer components (Nernst-Planck-Stokes). This is a clear departure from the molecule-centric hypothesis for conductive events. We would like to ask this reviewer: what is missing from the existing mathematical model which uses Poisson-Nernst-Planck equations to determine the flux imbalance for a specific pore geometry as a way to predict whether events will be conductive or resistive? In addition to this, the authors are unaware of any existing mathematical model that can predict conductive events in low salt conditions (for protein or PEG), or asymmetric salt (for protein and DNA) for which we can compare our predictions. Although there is a COMSOL model provided by Zhang et al., which models asymmetric salt conditions, the authors have major concerns about the model since the DNA is a smooth stationary rod which is far from accurate. The authors have explored adding DNA to COMSOL simulations over the past ~5 years and the results were always questionable in the viewpoint of the authors.

Manuscript Changes: We believe some of the additions below may further provide some credibility to the proposed hypothesis. We mainly aimed to provide some clarity to our main hypothesis and improve the discussion of the results. We also added data to the Supporting Information below:

Supporting Figure S3

Figure S3: Transferrin translocations through a 30 nm diameter pore under low ionic strength conditions (10 mM KCl buffered with 1 mM PBS) at two different pH values. Transferrin concentration was 350 nM and events were recorded at -400 mV. Under both pH values (one at isoelectric point and one above), conductive events were observed.

Main Text Changes:

Our experimental observations with PEG (a natively neutral polymer that functions as a polycationic polymer through cation adsorption) cannot be explained through this model where CEs were seen with smaller diameter pores where EOF is thought to be meager (Supporting Information Section 1). Additionally, protein (transferrin) translocation under low ionic strength condition, yielded CEs as well. (Supplementary Information Section 2).

The authors may find the following detailed comments on the manuscript useful when preparing the revisions. Abstract has too many abbreviations to be readable as a standalone summary of the work.

Response to Reviewer:

The abstract contains two abbreviations, EOF (electroosmotic flow) and CE (conductive event). Both were changed to to help readability.

Manuscript Changes:

Nanopore sensing is nearly synonymous with resistive pulse sensing due to the characteristic reduction of ionic flux during molecular occupancy of a pore, particularly at high

salt concentrations. However, conductive pulses are widely reported at low salt conditions wherein electroosmotic flow can be quite significant. Aside from transporting molecules like DNA, we investigated whether electroosmotic flow has other potential impacts on sensing attributes such current enhancements due to the analyte molecule. The overwhelming majority of literature reports counterions as the dominant mechanism of conductive events (a molecule-centric theory for conductive events). Conductive events are not well understood due to the complex interplay between (charged) nanopore walls, DNA grooves, ion mobility, and counterion clouds. Yet, the prevailing consensus of counterions being introduced into the pore by the molecule does not fit well with a growing number of experiments including the fact that proteins can generate conductive events despite having a heterogeneous surface charge. Herein, we demonstrate theory and experiments underpinning the translocation mechanism (i.e., electroosmosis or electrophoresis), pulse direction (i.e., conductive or resistive) and shape (e.g., monophasic or biphasic) through fine control of chemical, physical, and electronic parameters. Results from these studies predict strong electroosmotic pumping plays a role in driving DNA events and generating conductive events due to polarization effects (i.e. a pore-centric theory). We believe these findings will stimulate a useful discussion on the nature of conductive events and their impact on molecular sensing in nanoscale pores.

Line 49: Does CE stand for “current enhancing” or conductive event”? Probably the former, but writing is confusing.

Response to Reviewer:

“CE” is an abbreviation we have chosen to use which corresponds to a conductive event. In line 49, we remind the reader that current-enhancing events are synonymous with conductive events and provide our given abbreviation. The sentence has been slightly modified to help alleviate the confusion.

Manuscript Changes:

Therefore, pulses generated through translocations can be categorized as either current-reducing (i.e., resistive event, RE), or current-enhancing (i.e., conductive event: CE).

Line 50-68. I think the authors have misinterpreted the literature. There are no two separate hypotheses about two separate mechanism, i.e., an increase of ion concentration or effects of friction. Both are correct and occur at the same time, as described in Ref 3. That seems to be the consensus in the field.

Response to Reviewer:

The reviewer is correct although it seems that some literature emphasize specific aspects of the theory. The paragraph has been changed to retract the sentences referring to two different hypotheses and instead discusses them as a whole.

Manuscript Changes:

Despite the large number of experiments describing CEs, the origins of CEs in the presence of low ionic strength have been elusive. The leading consensus is that the combination of additional counterions and frictional effects influence the production of CEs³. The former describes how CEs stemming from low ionic strength conditions occur because the introduction of additional counterions by the charged DNA (i.e., K⁺) within the nanopore is greater than the number of ions within the DNA-free pore⁷. Once electrolyte concentration decreases below ~0.02 M, mostly counterions are present within the pore, which explains the current enhancement^{17,18}. Interestingly, at ~0.4 M, counterions are thought to precisely compensate for the DNA-occupied regions of the pore and yields no current modulation¹⁹. The latter relies on frictional forces (i.e., ionic friction with the grooves on DNA) which, in combination with the former, generate CEs^{3,9}. Although both predict the well-known crossover point in which events transition from resistive to conductive (via decreasing salt concentrations), the cation-specific, voltage-specific, and pore size-specific dependence of CEs have not been studied.

Lines 69-84. The logic is hard to follow. Should the reader associate anion and cation-selective conditions with REs and CEs? That's not obvious. Also, please also define what "flux imbalance" and "imbalance of ionic fluxes" (line 99) mean and how it is different from "ions selectivity".

Response to Reviewer:

In this section, we aim to introduce the idea that a flux imbalance will have an effect of generating either resistive or conductive events. The following text and figures will support this hypothesis and, later in the manuscript, we clarify the difference between ion selectivity and flux imbalances. Briefly, we define flux imbalance as the pumping of either anions or cations into the pore (as $|K^+ \text{ flux}|$ minus $|Cl^- \text{ flux}|$). This can be toggled by manipulating the pore size, salt concentration, voltage applied, or salt type used. This differs from ion selectivity in that ion selectivity is commonly a characteristic of the pore properties itself. Flux imbalances can be produced through external changes. We have provided additional information in regards to the definition of flux imbalance as well as the difference between flux imbalance and ion selectivity into the Supporting Information Section 8.

Manuscript Changes:

"Flux imbalance, defined here as $|K^+ \text{ flux}|$ minus $|Cl^- \text{ flux}|$, can be generated through externally applied conditions and parameters; for example, flux imbalance increases with both the pore diameter and the applied voltage (Supporting Information Figure S11)."

Flux imbalances differ from ion selectivity in that a flux imbalance can be produced through externally manipulating the pore size, salt concentration, voltage applied, or salt type (i.e., KCl,

LiCl, CsCl) whereas ion selectivity stems from the properties of the pore itself. Under symmetric, low ionic strength conditions (10 mM KCl), EOF pumps cations into the pore, producing a flux imbalance in favor of K^+ , yielding CEs when DNA translocates the pore. The surrounding pore environment can be altered to have a cationic flux imbalance when a concentration gradient is used (1 M KCl inside and 4 M KCl outside). While a negative voltage is applied inside the pore, K^+ is pumped into the pore via EPF, producing CEs when DNA exits the pore. Similarly, the pore environment can have a flux imbalance in favor of anions, where 4 M KCl is inside the pore and 1 M KCl is outside. When a positive voltage is applied to attract DNA, Cl^- is pumped into the pore. Upon translocation, REs are generated. Thus, we conclude that a flux imbalance in favor of cations produces CEs and a flux imbalance in favor of anions produces REs.

Line 87: The EOF-driven transport of DNA was first predicted in *J. Phys.: Condens. Matter* 22 (2010) 454123, well ahead of the cited references. Please correct this omission.

Response to Reviewer:

This omission has been corrected and we have changed the sentence to mention the first prediction of EOF-driven DNA transport (*J. Phys.: Condens. Matter* 22 (2010) 454123).

Manuscript Changes:

Although it may seem ostensibly obvious, remarkably, electroosmotic dominant transport of DNA is hardly reported (first predicted in 2010²¹) and therefore, less known in the nanopore community^{22,23}.

Figure 1 needs a schematic defining the polarity of the bias, the direction of the EPF and EOF forces, the direction of the ionic current and EOF with respect to the pore geometry.

Response to Reviewer:

For Figure 1 (low ionic strength conditions), we believe it is most important to label the directionality of EPF and EOF and have included a schematic in Figure 1b. As for pore geometry dependence and polarity bias, more information can be found in Figure 3 as well as the Supporting Information in regards to these inquiries. We feel adding more schematics to Figure 1 will be distracting to most readers and we wish to keep pore geometry and polarity bias information separate from Figure 1's information.

Figure 1b: please specify which part of the dependence was used for the linear fit and Eq. 1

Response to Reviewer:

Clarification has been added to address this comment. Specifically, we have used the linear portion at the negative applied voltages to estimate our pore diameters when TEM was not available. This is mainly because: (1) under low ionic strength conditions with negatively applied voltages, DNA translocates into the pore and (2) rectification was present during each of our I-V curves. Therefore, using the negative voltages to calculate the pore diameter gives us the most accurate linear slope and pore size estimation (as negative voltages are utilized in these experiments).

Manuscript Changes:

The G, measured by calculating the slope of the linear portion at the negative voltages, varied between 0.58 and 5.35 nS and the I-V curve showed ionic current rectification which is consistent with the previous reports³².

Caption Changes:

I-V curves pertaining to four differently sized nanopipette orifices. For pore size estimations, the linear portion at the negative voltages was used (yellow shaded region). The schematic within the I-V curves shows the directionality of EOF and EPF at negative voltages.

Figure 1c is incomprehensible. Please draw a schematic of the continuum model defining what “the distance to pore” is. Is it along the capillary or radial? If the former, what was the pore size, cone angle, how do the values depend on the two? One can add one set of extra traces without complicating the plot. Also, the fluid velocity is not uniform and the DNA occupies regions of different flow magnitude. What happens then to the molecule, does it stretch, breaks? Different parts of the molecule cannot move with different velocities for very long.

Response to Reviewer:

We have improved upon Figure 1d to display the axis of symmetry (gray line). This was done to also help clarify the distance from the pore figure (Figure 1c) is to be taken from the gray line.

Fluid velocity can unfold DNA because of these different flow velocities shown at the tip in Figure 1c. This typically results in stretching the molecule. DNA’s net velocity is most sensitive to the surface charge of the pore as well as the salt concentration in the pore and in the bath. We have added a few statements within the Supporting Information Section 5 to describe this feature. As shown by others, the DNA can be under tension or compression based on the translocation direction. (Nature Communications volume 8, Article number: 380 (2017)) In our case, the DNA outside the nanopipette would be in compression. The DNA coil dynamics should be nearly identical to voltage-driven translocations since they electric field also is not uniform and exponentially increase as one approaches the pore.

Manuscript Changes:

The local velocities of DNA, as it translocates through the pore, fluctuate throughout the length of the molecule⁶. In Figure 1c, the net velocity of DNA is in favor of EOF and remains below 0.2 m/s throughout all distances from the pore. It can be said that because EPF is in opposition to EOF at negative applied voltages, the translocating DNA molecule can become linearized/unfolded. Additionally, the DNA molecule can become stretched due to the differing flow velocities at the pore tip. The net velocity of DNA is mostly affected by the surface charge on the pore as well as the salt concentrations within the pore and bath.

Line 156: Define what “operational conditions adopted in this work” are. The prior description lists a range of mutually exclusive situations.

Response to Reviewer:

We have clarified in the manuscript that “operational conditions adopted in this work” refers to low ionic strength conditions.

Manuscript Changes:

Herein, we adopted the operational configuration where the anode electrode is placed inside the pipette side and grounded electrode in the bath (under low salt conditions).

Figure 1d. What is “pump velocity”? Is the pump moving somewhere? What is the meaning of the white lines? What is actually shown in the inset? The caption says fluid velocity, but it appears to be an image of something labeled.

Response to Reviewer:

EOF creates a flow profile introducing ions in the bulk to the pore interior and, eventually, out of it. Essentially, EOF pumps ions from one side of the nanopipette to the other. Pump velocity is describing the rate in which this occurs.

The white lines located in Figure 1d are fluid flow lines. Under low ionic strength conditions and at an applied negative voltage, the fluid flow lines are hugging the outer nanopore walls, directing DNA down towards the pore aperture. As for the inset, we have now properly labeled and described the reason for showing this image. Briefly, it is a z-stacked time series of images of DNA labeled with YOYO-1. With a voltage of -700 mV applied, DNA translocated through the pore via EOF and the resulting image indicates the pathway that the molecule took. The path generated by DNA entering the pore as seen through image stacking is vastly similar to the simulation of fluid flow lines in Figure 1d.

Manuscript Changes:

Given the inherent differences associated with capture volume shapes associated with EOF and EPF dominant mechanisms, the next step was to elucidate the entrance trajectory of DNA. To do this, λ -DNA was added to the bath and a negative voltage bias was applied to the other electrode to ensure if translocations were to happen (i.e., from the bath to the tip side; forward translocation direction), it would be caused by electroosmosis rather than by the conventional electrophoresis. The fluid flow profiles around pore-tip were simulated to further understand the EOF-driven capture of DNA. The simulated results are shown in Figure 1d and indicate DNA proceeds to diffuse around the solution until it enters the EOF capture volume, where it is then transported through the pore. To reiterate, this transport is fundamentally possible when the EOF velocity is greater than the EPF drift velocity. Since DNA events occur anti-EPF, mapping the fluid motion is indicative of the capture zone. To experimentally validate the finite element analysis (Figure 1d), λ -DNA was tagged with YOYO-1 and the nanopipette tip placed in the focal plane of a water immersion objective (Nikon, NA=1.2). A stacked time series of images (acquired from a Princeton Instruments ProEM EMCCD) allowed us to observe λ -DNA capture at -700 mV (Figure 1d inset reveals that fluid motion along the sides of the pore is responsible for λ -DNA translocation).

Figure 1 is missing several examples of current blockades (or enhancements) associated with the EOF-driven translocation events.

Response to Reviewer:

Although there aren't any current event traces found in Figure 1, we have included three current traces of the three different conditions tested in Figure 2. Additionally, current traces can also be found within the Supplemental Information.

Consider flipping the orientation of Figure 1d to be consistent with 1e.

Response to Reviewer:

The orientation of Figure 1d and 1e are now the same.

Manuscript Changes:

See above for the updated Figure 1.

The data shown in Figure 1e are really interesting! Still, this reviewer finds it very surprising that the capture volume extends to millimeters. Are there any theoretical calculations that would support such a macroscopic range, which is three orders of magnitude larger than that for EP capture?

Response to Reviewer:

As shown in Figure 1, the model that we choose to explain DNA transport (by EOF) was using the electrophoretic drift velocity of DNA and comparing that to the EOF flow velocity. Since both provide units of velocity (m/s), they can be summed to find the net velocity. Interestingly, the electric field is much more geometrically confined and leads to the electrophoretic force decreasing much faster (especially outside the nanopipette). The authors speculate that the conservation of mass (since EOF acts on the fluid, rather than electrophoresis only acting on the charged species) and fluid dynamics promote a larger capture (i.e., if the pore is pumping fluid into the pore, fluid has to flow in from elsewhere in the bath). EOF capture is thus more like DNA getting caught in a “river-like” current (fluidic current, not ionic current). In this case, even small drift over the mm-scale can eventually provide a constant delivery of DNA to the pore. Whereas electrophoretic capture is typically thought to be all-or-nothing. The authors believe the fluid-flow delivery of DNA to the tip is thus mechanistically different. We expanded the COMSOL simulation to show where the DNA is coming from within the flow cell.

Have the authors ruled out that what is seen in Figure 1e relates to capillary pressure or partially filled pipette? Why not submerge the entire capillary in the fluid?

Response to Reviewer:

We can confidently say that this observation is not due to capillary pressure or a partially filled pipette. When performing experiments, the placement of the electrodes and the nanopore itself does not allow for a build up of pressure to accumulate at any location. Prior to conducting experiments, the nanopipettes are visualized under an optical microscope for any breaks, clogs, or air bubbles that will affect the current recorded. Only until after the nanopipette has passed this examination will it be used to record translocations.

The authors do not anticipate that the capture rate would be influenced very much, if any at all, upon submersion of the entire capillary into the electrolyte solution. While there is a significant jump in event frequency from 1.0 to 4.2 mm (1.25 to 40 Hz), we do not expect the frequency to increase much more than 40 Hz. For all other low salt experiments performed (~20 trials), the entire taper was submerged into the electrolyte solution.

We have included these expectations into the Supporting Information Section 6.

Manuscript Changes:

It is important to note that each nanopipette was examined using an optical microscope prior to any experiments. This was performed to ensure the pore was free of any breaks, clogs, or air bubbles that may affect the quality of the current fluctuations. Our largest depth was recorded at 4.2 mm, meaning that the entire taper was submerged within the electrolyte solution. We speculate that the event frequency will not increase vastly above 40 Hz for depths exceeding the entire taper (i.e., the entire capillary), as that would entail the EOF capture zone to extend into the centimeter regime. Further validation of this model is needed but may provide a framework for future experiments.

Line 180: "...nanopore was suspended from a micrometer" What does this mean?

Response to Reviewer:

The nanopore was secured (mounted) onto a manual linear stage actuator. The linear stage can be lowered or raised by twisting the mechanical knob. By identifying the location where the pore tip barely makes contact with the top of the electrolyte solution (this was done by applying a small voltage and then lowering the pore until the circuit was complete, i.e. the current changed), we can then lower the nanopore to different depths.

Manuscript Changes:

For exact measurements, the nanopore was suspended from a linear stage containing a linear actuator.

Figure S6 Caption:

The nanopore was mounted onto a manual linear stage actuator. The stage was raised and lowered by twisting the mechanical knob accordingly. Depth zero was identified by applying a small voltage into the pore and lowering the nanopore closer to the electrolyte bath. Once the current changed (in response to the circuit becoming connected), that location was denoted as the depth at 0 mm. From there, the mechanical knob was twisted to yield the following depths: 0.53, 1.1, and 4.2 mm (the entire taper).

SI does not seem to be organized to follow the flow of the main text.

Response to Reviewer:

Thank you for bringing this to our attention. We have since modified the order of the Supporting Information to match the manuscript so now both are in agreement.

It is not clear why the following is assumed to be true, please elaborate or better provide some simple mathematical arguments: “If counterions become sheared off the negatively charged DNA or overlapped with the counterions from the pore wall to enhance ion concentrations inside the pore, CE amplitude should increase with decreasing pore size”.

Response to Reviewer:

This raises a good point. We don't believe this sentence adds much substantial information and thus, it has been removed.

Manuscript Changes:

Finally, using a custom-coded MATLAB script, translocation conformations of DNA were examined which revealed that DNA adopts the widely seen conformations: linear, partially folded and fully folded (see Supporting Information section 6 for further details). By solely selecting linear events, we were able to evaluate the relationship between CE amplitude and pore size; a relationship that may be hidden by multiple conformations of DNA. As seen in Figure 1f, no observable trends were seen in CE amplitude with pore conductance (a proxy for pore size).

Trace in Figure 2a is the same as in S7. Reusing the trace leaves an impression that that was the only trace collected.

Response to Reviewer:

Figure trace in Supporting Information has been changed to reflect that more than one trace was collected.

Manuscript Changes:

Figure S9: Event Classification of DNA. A DNA molecule would translocate (a) linearly, (b) partially folded or (c) full-folded. Other conformations were negligible. Events concatenated together. Red lines are guides for the eye to recognize three different DNA configurations.

Figure 2 “Observations on the nature of events ...” sounds very esoteric. Is it possible to have a more straightforward title? Just describe the experiment. In general, the word “Observations” cannot describe what is shown, it is a process.

Response to Reviewer:

Figure 2’s caption has been modified to briefly describe the details presented.

Manuscript Changes:

Figure 2: Event properties of DNA under low ionic strength and asymmetric salt conditions. (a) Typical event structures observed with λ -DNA translocation experiments under low salt, symmetric conditions at -600 mV. The three events correspond to linear, partially folded, and fully folded λ -DNA from left to right. The schematic on the right displays the salt conditions as well as the voltage applied to either side (denoted by positive or negative signs) and how λ -DNA enters via EOF (anti-EPF) from the capture zone located at the outer walls of the nanopore. (b) Observation of CEs in asymmetric salt conditions when λ -DNA + 1 M KCl was added into the pipette and 4 M KCl was outside at -600 mV. To the right, EPF is used to repel λ -DNA away from the negatively applied voltage and exit the pore into the bath solution. (c) Current traces of REs in asymmetric salt conditions when λ -DNA + 4 M KCl was added into the bath and the pore contained 1 M KCl. Located to the right is a schematic showing how DNA is electrophoretically attracted to translocate into the pore when $+600$ mV is applied. All buffers were prepared at pH 7.4. (d) Event decay to equilibrium for the case experiments shown in (b). DNA is exiting the

pore and thus should immediately leave the sensing zone of the pore as opposed to the reverse translocation direction.

Back in 2009, all-atom MD simulations have shown that current enhancement can be seen in high molarity experiments, see Figures 11 and 12 of *Biophysical Journal* 96:593-608 (2009). Pore geometry and non-homogeneous ion distributions can affect the blockade level in a very non-trivial way. The authors should discuss their finding in the context of that work.

Response to Reviewer:

We do agree with the statement that pore geometry and non-homogenous ion distributions can affect the current change. However, the pores simulated in their findings was denoted to have diameters ranging from 1.8 to 2.1 nm, which are vastly smaller than all the pores used in our study. As such, when DNA translocated in their simulations, it began to gather on the trans side (primarily once the dsDNA entered the pore), increasing the concentration of ions and thus creating a current enhancement. In our experimental conditions, with pores ranging from 9 to 50 nm in diameter, it is not energetically favorable for DNA to gather upon translocation, which is why we see few knots. The majority of DNA in this study translocate either linearly, partially folded, or fully folded. Additionally, the gathering witnessed in the simulations seems to have come about due to dsDNA translocating the pore (most likely due to the small pore diameter hugging the molecule). This would not occur under our conditions because the pore diameters are at a minimum 3x larger than the width of DNA.

We are in agreement with what is stated in Figure 12. DNA will always block a portion of the ionic flow within the pore as it translocates. Then, depending on the electrolyte concentration, it is possible that the DNA counterions outweigh the ions that are blocked, resulting in a CE or, the counterions do not outweigh the blockage and the resulting event is resistive. While the authors agree with CEs not being limited to low salt conditions only, there are many differences between the simulation parameters and our experimental set-up. Therefore, we will mention the discovery of Comer et. al. within the manuscript.

Manuscript Changes:

Simulations performed in 2009 predicted that current enhancements could be seen at high ionic strength conditions with small pore diameters (<2.2 nm) using hair pin DNA. In acknowledgment of that finding, we also show that CE phenomenon is not limited to low ionic strength conditions.

Line 206: What exactly is “directional dependence of DNA transport”? That DNA can move in both directions when one switches polarity of the field? Probably not what the authors meant, but what did they mean?

Response to Reviewer:

There was an error within this section. Thank you for bringing that to our attention. It has been changed to describe the voltage in terms of EPF-dominated transport. In other words, either attracting or repelling DNA in the desired direction.

Manuscript Changes:

To show that CE phenomenon is not limited to low ionic strength configuration, we used salt concentration gradients where the pipette was filled with 1 M KCl and the bath was filled with 4 M KCl. λ -DNA was either added to the pipette (Figure 2b: case I) or bath (Figure 2c: case II) and a voltage bias consistent with the conventional EPF-dominated transport was applied to the pipette.

Line 207 “a change in the conductive nature of the pulses” ... confusing, better use simple words like a transition from RE to CE upon reversal of bias polarity or something similar.

Response to Reviewer:

To make it easier for readers to understand, we have replaced “conductive nature” with “direction”. With this, we believe this statement has been made less confusing.

Manuscript Changes:

Although directional dependence of DNA transport has been reported previously with nanopipettes⁴⁵, a change in the direction of the pulses has not been previously observed.

Line 212 “produced anti-conventional event shapes” What is opposite of conventional? Unconventional? In what way? Perhaps the authors should show both “conventional” and “anti-conventional” shapes to give a reader a chance of grasping the meaning. Why is that important?

Response to Reviewer:

Yes, in this section we accidentally used the word “anti-conventional” instead of unconventional. This has been changed. We use this term (unconventional) to describe how the events differ from what is typically seen given the direction in which DNA is translocating. What we are referring to can be seen in Figure 2b and c, specifically within the zoomed-in portions of the current trace. When DNA exits the pore, usually the current trace is similar to a square pulse. On the other hand, when DNA enters the pore, usually the event shape contains a tail at the end because it travels along the confined taper region. However, in both cases, this is not what we observe. We witness the exact opposite wherein DNA entering the pore yields square-like pulses and exiting the pore, the events contain tails. We have included additional text to encourage the reader to look at the current traces provided in Figure 2b and c for examples of conventional and

unconventional event shapes. Lastly, we have made a few formatting changes within the figure to aid in grasping the meaning of these terms and their associated shapes.

Manuscript Changes:

Figure 2b red inset and 2c blue inset provide examples of unconventional shapes seen under their respective conditions.

Caption Changes:

(b) Observation of CEs in asymmetric salt conditions when λ -DNA + 1 M KCl was added into the pipette and 4 M KCl was outside at -600 mV. To the right, EPF is used to repel λ -DNA away from the negatively applied voltage and exit the pore into the bath solution. Red inset: DNA exiting the pore produces CEs containing a “tail” before returning back to baseline. (c) Current traces of REs in asymmetric salt conditions when λ -DNA + 4 M KCl was added into the bath and the pore contained 1 M KCl. Located to the right is a schematic showing how DNA is electrophoretically attracted to translocate into the pore when +600 mV is applied. All buffers were prepared at pH 7.4. Blue inset: DNA entering the pore yields REs that look similar to square pulses.

Figure 3 and the rest of the manuscript. Unfortunately, the authors never defined what “flux imbalance” means, which makes the rest of the manuscript difficult to understand. Is it the same as ion selectivity, which is the difference in the magnitude of the currents carried by different ionic species? If so, it is not at all surprising that moving fluid will change the contribution of ionic species to the total current. Or does it refer to some transient, non-equilibrium phenomena leading to ion cloud polarization because of the finite volume of the system? How does one transform the latter into a quantitative prediction about the magnitude of the ionic current blockade? If that is the case, the finite size of the volume must be explicitly taken into account.

Response to Reviewer:

We have taken this suggestion and included an additional section within the Supporting Information Section 7 that discusses the definition of a flux imbalance as it applies to our manuscript as well as the difference between a flux imbalance and ion selectivity.

Under symmetric, low salt conditions where DNA is in the electrolyte bath, cations are pumped into the pore upon application of a negative voltage. This electroosmotic flow then allows DNA to translocate, giving rise to CEs. Under asymmetric, high salt conditions where DNA is in the pore, cations are pumped into the pore once a negative voltage is applied, and CEs arise. In the last condition where DNA is in the 4 M KCl bath solution, a flux imbalance in favor of chloride is produced when a positive voltage is applied and REs are seen as a result.

Manuscript Changes:

Flux imbalances differ from ion selectivity in that a flux imbalance can be produced through externally manipulating the pore size, salt concentration, voltage applied, or salt type (i.e., KCl, LiCl, CsCl) whereas ion selectivity stems from the properties of the pore itself. Under symmetric, low ionic strength conditions (10 mM KCl), EOF pumps cations into the pore, producing a flux imbalance in favor of K^+ , yielding CEs when DNA translocates the pore. The surrounding pore environment can be altered to have a cationic flux imbalance when a

concentration gradient is used (1 M KCl inside and 4 M KCl outside). While a negative voltage is applied inside the pore, K^+ is pumped into the pore via EPF, producing CEs when DNA exits the pore. Similarly, the pore environment can have a flux imbalance in favor of anions, where 4 M KCl is inside the pore and 1 M KCl is outside. When a positive voltage is applied to attract DNA, Cl^- is pumped into the pore. Upon translocation, REs are generated. Thus, we conclude that a flux imbalance in favor of cations produces CEs and a flux imbalance in favor of anions produces REs.

Reviewer #2 (Remarks to the Author):

The manuscript by Lastra et al. reports on new, detailed studies of DNA and protein translocation through nanochannels. In particular, they investigate the role of electroosmotic flow (EOF) and electrophoretic (EP) transport on the direction of transport, both experimentally and via simulations. While this has been studied to some level of detail for proteins, the authors correctly point out that for DNA this is severely understudied. Indeed, one key finding of the study is that also in the case of DNA the direction of transport can be determined by EOF, rather than EP. Interestingly, the authors then also go on to propose a new model for how the event characteristics emerge, highlighting the importance of polarisation effects during temporary channel blockage. Concentration polarisation is of course a common phenomenon and routinely considered in membrane science, but it is indeed new and very interesting in the present context. This is because our physical understanding of the origin of resistive/conductive pulse sensing is closely linked with the interpretation of the results and, ultimately, also to the capabilities of the sensor. For example, it provides a new perspective on the interpretation of the relation between translocation time and DNA length, the spatiotemporal resolution of the sensor and how sub-structure in DNA are detected (presumably when polarisation effects are minimised?). It is becoming clear that the actual current-time events are a convolution of multiple effects, depending on the conditions, which may include volume exclusion, ion condensation and, according to the authors, also concentration polarisation. Accordingly, the authors present a range of arguments in support of the model, which warrants further investigation.

Formally, the work is of a high standard throughout. The text is generally written well, even though there are some typographic errors and misprints (p. 4, l. 74 "stokes" should be capitalised; p. 7, l. 137 "oriface" or l. 141 "aparature"; p. 17, l. 297 "Res" should be "REs", several on p. 18 and so on). Some more careful proof-reading is advised before submitting a revised version of the manuscript. On several occasions, references are also missing, for example on p. 6, l. 130 (where does the value for the surface charge come from?) or on p. 7, l. 155 (the value for the electrophoretic mobility of DNA). The terminology around which electrode is anode and cathode needs to be corrected (p. 7, ll. 143), since also the grounded electrode will act as the respective other (depending on the bias applied to the other electrode).

Response to Reviewer:

Thank you for providing feedback in regards to our manuscript. We have adjusted the manuscript to address all the errors mentioned in the above comment. Additionally, we have supplemented the manuscript with two added references to cite the values of the glass surface charge as well as the electrophoretic mobility of DNA.

For the section in which the anode and cathode are mentioned (page 7), this description has been changed to better convey our experimental set-up. Specifically, in this section, we are discussing Figure 1, which is describing our experiments under low ionic strength conditions. In this scenario, the anode is placed inside the nanopipette and the ground electrode within the electrolyte solution. This is not changed for any other data presented in this paper under low ionic strength conditions. Also, we have dropped the word “conventional” as this may be a bit relative.

Manuscript Changes:

Herein, we adopted the operational configuration where the anode electrode is placed inside the pipette side and grounded electrode in the bath (under low salt conditions).

Overall, I think this work should be of great interest to the community and, subject to minor revisions, can be considered for publication in the journal. It clearly gives new impulses to the field and highlights the potentially more general importance of "dynamic" effects in the context of nanopore sensing.

Reviewer #3 (Remarks to the Author):

The authors report a very exciting study on glass capillary nanopores, a very interesting technique to probe single molecules (e.g. DNA) and to study Physics at the Nanometer Scale. Compared to many other (cited) works, they apply a low salt concentration in the buffer solution which enhances electrostatic effects. For the first time (to my knowledge) they systematically study transport of DNA molecules and fluid caused by a salt concentration gradient. The experimental work is accompanied by simulations based on the electrokinetic equations, a set of partial differential equations describing the coupled interaction and motion of ions and the surrounding fluid.

The applied methodology is sound and in line the state of the art (to my knowledge). The rich complexity of the physics leads to an interesting spectrum of observed conducting and resistive events (CE/RE) where the presence of a DNA molecule in the pore modulates the current towards higher or lower values. The works clearly show that the methodology is well suited to gain understanding of the physics at that length scale. The authors argue that the flux imbalance, i.e. asymmetric contribution of both ion species to the conductivity due to electrostatic exclusion

of one ion species is a key factor to explain the observations. I am very supportive of the author's idea to disentangle the different physical effects, and to find intuitive explanations for this (to my knowledge) still unsolved puzzle. But the author's conclusion that flux imbalance explains most the observations is too short. A combination of flux imbalance with other effects (e.g. electroosmotic flow) is required to cause the effect. Furthermore, it is important to stress, that many of the observed effects are effects non-linear in the applied voltage (e.g. the inversion of the event in Fig 4a), and this should be discussed in more detail.

Response to Reviewer:

We agree with the reviewer. The main message of this manuscript is not to conclude that flux imbalance is the primary hypothesis for witnessing either conductive or resistive events. We aim to include the imbalance of fluxes in the answer to the question “Why does ionic current *increase* during transient DNA occupancy of a nanopore?”

In Supporting Information Figure S5, we display the median current changes as a function of applied voltage for 10 mM LiCl. For this figure we have five voltages: -300, -500, -600, -700, and -900 mV. Because events seen at -600 mV were biphasic and nearly equiphase, we analyzed this data twice, once looking at the conductive amplitude and once looking at the resistive amplitude. For the CEs (-600, -700, and -900 mV) we see a mostly linear relationship between current amplitude and voltage applied. The same observation can be made for the REs occurring at -300 and -500 mV. To conclude, the change in current with respect to voltage seems to be mostly linear with the exception of a polarity change occurring at -600 mV. This discussion has now been included into Supporting Information Figure S5.

Manuscript Changes:

(b) Events stemming from 10 mM LiCl are further separated by CEs or REs. For all voltages, the RMS noise value was 7 ± 1 pA, suggesting that the main cause of difference in SNR is the change in current. We see a mostly linear relationship between the change in current and voltage applied with the exception of a polarity change occurring at -600 mV. Specifically, the CE amplitudes seen at -600 mV, -700 mV, and -900 mV linearly increase as the voltage increases in negativity. Likewise, we witness the same observation for REs occurring at -300 and -500 mV. The change in current with respect to voltage applied is mostly linear with the exception of a polarity change which occurs at -600 mV.

From my perspective the (difficult) theoretical analysis is a weakness of the manuscript. From my point of view, this is reflected in the fact that a few aspects and concepts are missing. First, this includes the discussion of linear vs. nonlinear response, and the respective perturbation approaches in the literature. Furthermore, this includes the following points.

Response to Reviewer:

The linear vs. non-linear response has been updated and included into the Supporting Information as stated above. Non-linear perturbations in the relationship between current amplitude and voltage applied could also be indicative of different DNA configurations entering the pore (Chen et al, NanoLetters 2004), We have also included this reference in our discussion of linear vs. non-linear current amplitude responses to voltage.

Manuscript Changes:

Lastly, for this figure, we have only considered linearly translocating DNA. It is unlikely, but there is a small chance that DNA configurations other than linear were not excluded properly with the MATLAB analysis. Because DNA linearization has some dependence on voltage⁶, it could possibly explain the non-linear relationship mostly occurring at -600 mV.

The authors introduce the concepts of electroosmosis and electrophoresis well, but do not mention the main quantity which is typically used to quantify the (effective) magnitude of EOF and electrophoretic mobility, the zeta potential. It unites the concepts of electrophoresis and electroosmosis, which is very useful as it is governed by identical physics. In their comparison, the authors compare the electrophoretic mobility of DNA which includes an effective surface charge and a glass charge density where it is not clear if this how this quantity was measured.

The other fundamental quantity that is not introduced is the Debye length which describes the range of interactions between a charged surface and the surrounding solution. The experiments are very interesting as, other than many works, the low salt regime leads to a situation where the Debye length is comparable to the pore diameter leading to conditions where direct electrostatic and electrokinetic interactions of the DNA with the pore become relevant.

Response to Reviewer:

Because the zeta potential and Debye length are closely related, we decided to combine the reviewers feedback into one response. In our simulations, we have used values of the surface charge of silica glass as well as the electrophoretic mobility of DNA taken from previous literature (references 33 and 40, respectively). Additionally, the ionic diffusion coefficients and electrophoretic mobilities used in our simulations provide sufficient basic transport properties. However, they lack specifics such as the geometric size of the ions. Because of this, and to better understand the connection between electro-hydrodynamics and the Debye layer screening of the pore surface, streaming current measurements were performed and the results can be seen in Supporting Information Figure S12.

Although both are not mentioned within the main text, we do discuss both within the Supporting Information. We also describe how streaming currents (KCl, LiCl, and CsCl) relate to the Debye layer as well as the zeta potential in Supporting Information Section 9.

Manuscript Changes:

When an electrolyte solution is traveling through the negatively charged glass nanopore via pressure, an electrical double layer is constructed. In this layer, there is an increase in cation (K^+ , Li^+ , or Cs^+) concentration and a decrease in chloride concentration locally at the pore surface. The resulting electrical double layer produced by streaming currents is comparable to the Debye screening layer and the streaming potential is directly related to the zeta potential¹⁰. Thus, by performing streaming current measurements, we obtain information relating to the Debye length and zeta potential at low ionic strength conditions.

Furthermore, the authors perform very interesting experiments with asymmetric salt conditions, which gives rise to osmosis, where diffusion ions draw fluid along through the pore. The authors do not consider this effect even though it appears to me to be not very difficult to include into the simulation as only boundary conditions need to be altered. Then, the resulting fluid flow magnitude could be compared to the electroosmotic result indicating how this affects the fluid flow profiles.

Response to Reviewer:

While this is a valid suggestion, the authors are unaware of any existing boundary condition alterations that can be done to simulate osmosis. For example the only boundary condition that is relevant and capable of generating flow is the pressure at the inlet and outlet which is not straightforward to predict or correlate to osmosis. We also found no analytical equations to calculate the osmotic pressure for our experimental conditions. We also need to consider if a boundary condition is the correct approach to model osmotic flow; for example, as opposed to a “volume force” acting on the liquid (which is done for EOF flow). For asymmetric salt conditions, we aim to emphasize the flux imbalances between cations and anions give rise to CEs and REs, respectively, and how it is possible to toggle between the two by manipulating certain external parameters. Simulating the fluid velocity profiles under asymmetric salt conditions as well as osmosis will be a great addition to a future manuscript.

The observed tails in conductive events are very exciting as they indicate that "something nonlinear" is happening. As the tails only appear when the osmosis opposes the electrophoretic DNA transport, I would speculate that the DNA does not leave the pore orifice region as fast as when osmosis is not present. The authors speculate that it is a transient effect of the electrokinetics. An upper bound for the time scale of relaxation from the diffusion constant of the ions ($\sim 10^{-9} \text{ m}^2/\text{s}$) and the relevant length scale of the distortions (which is difficult to pick carefully). Assuming a relevant length scale of 100 nanometer, I obtain a relaxation time of 10 microseconds. Transient effect of the fluid are governed by the kinematic viscosity ($\sim 10^{-6}$), which leads to an even faster time scale. Therefore I have doubts regarding this explanation.

Response to Reviewer:

Under asymmetric salt conditions (1 M KCl inside the pore and 4 M KCl outside), there is a large increase in charge density that occurs in close proximity to the pore. This charge that is stored at the conical nanopore tip is voltage dependent (as seen in Figure 3f). To provide more information, we performed time dependent modelling to determine the time it takes for the charge to dissipate when voltage is removed. This occurs in a few microseconds. We have provided additional information surrounding this simulation as well as describing the observed tails within the Supporting Information Figure S6.

Manuscript Changes:

A new simulation figure has been inserted in Supporting Information Section 4.

COMSOL modelling demonstrating the timescale of charge dispersion once EOF pumping is removed (voltage bias: -600 mV). The timescale of charging and discharging accumulated charge is fast (3-5 μs to reach steady state space charge density).

In summary, the authors have performed very exciting work towards understanding the physical mechanisms. But I recommend reiterating the theoretical discussion. Maybe, from simulation find analogies, systematically study the transition from linear to non-linear response, etc this is possible - although I personally experienced it being difficult. I have named a few concepts I believe should appear in the discussion.

Response to Reviewer:

Thank you very much for your time and feedback. We have added a few points of discussion in regards to the linear vs. non-linear response into the Supporting Information as well as the Debye length and zeta potential that we hope is to your satisfaction.

Manuscript Changes:

“When an electrolyte solution is traveling through the negatively charged glass nanopore via pressure, an electrical double layer is constructed. In this layer, there is an increase in cation (K⁺, Li⁺, or Cs⁺) concentration and a decrease in chloride concentration locally at the pore surface. The resulting electrical double layer produced by streaming currents is comparable to the Debye screening layer and the streaming potential is directly related to the zeta potential¹⁰. Thus, by performing streaming current measurements, we obtain information relating to the Debye length and zeta potential at low ionic strength conditions.”

And finally two small remarks. The authors deduce the pore diameter from conductivity measurements. They state they infer the conductivity from the slope of the I-V curve. Obviously, the curve is nonlinear, and this makes me wonder how they have taken the slope. I think this is worth commenting on. This includes the error bars on pore diameters (l. 121). I believe that the nonlinearity of the curve violates the assumptions of the diameter formula (1) the curve still should create the right qualitative picture. As the authors confirm the diameter measurements with TEM, the method is valid.

Response to Reviewer:

The linear portion of the I-V curve was only taken into consideration when estimating pore diameters. Under these conditions, the linear portion occurred when applying negatively charged voltages; which is also where most translocations occur.

Manuscript Changes:

The G , measured by calculating the slope of the linear portion at the negative voltages, varied between 0.58 and 5.35 nS and the I-V curve showed ionic current rectification which is consistent with the previous reports³².

Caption Changes:

(b) I-V curves pertaining to four differently sized nanopipette orifices. For pore size estimations, the linear portion at the negative voltages was used (yellow shaded region). The schematic within the I-V curves shows the directionality of EOF and EPF at negative voltages.

l. 220 "tale" -> "tail"

Response to Reviewer:

This error has been corrected, thank you.

Peer review comments, second round review –

Reviewer #1 (Remarks to the Author):

The authors have addressed all points raised in the previous round of review.

Although the manuscript can be accepted as is, the authors are asked to consider a replacement to their frequently used language construct "to pump ions into a nanopore". To this reviewer, the construct implies that ion concentration continues to increase, as in "to pump water into a bucket". Perhaps an easy fix would be to replace "into" with "through", which does not imply accumulation.

Congratulations are in order for the authors on completing such a novel and exciting study

Reviewer #2 (Remarks to the Author):

I am satisfied that the authors have addressed all my concerns raised in connection with the previous version of the manuscript.

Reviewer #3 (Remarks to the Author):

I'd like to thank the authors for assessing my and the other reviewers' criticism and substantially improving the quality of the manuscript. This makes many experiments, parameters, concepts and ideas more clear. While the experimental part of the manuscript still is very impressive and especially the finding that identical asymmetric salt conditions, the sign of the observed events inverts when at the same time inverting the voltage and the DNA concentration. However I my arguments regarding the weakness of the theoretical part could not yet be ruled out. I think the complexity of the system requires a rigorous analysis and a very careful handling of the wide range of theoretical concepts.

In the theoretical analysis of electrokinetic phenomena, typically a linear expansion in the applied driving forces is performed (e.g. O'Brian & Wight Electrophoretic Mobility of a Spherical Colloidal Particle, 1978). Going beyond the linearity assumption is a strong but required step for understanding e.g. conduction rectification of nanopores. All effects that are not captured in the linear response require an especially thorough consideration and many intuitions and concepts from the "classical" theory are not valid any more. One example is the notion of conductance - the concept is based on the linear response to infinitesimal perturbation (=linear response theory, with a very sound apparatus of StatMech supporting this concept, Fluctuation-Dissipation theorem, reciprocal relations, etc). I don't believe that taking the slope of only the positive branch of the I-V curve is adequate. Flux imbalance in a system with weak boundary effects due to small Debye length clearly is a nonlinear phenomenon. It can only occur as an effect of strong applied driving forces. Being itself a result of strong nonlinearity, using the concept of flux imbalance as a *cause* rather than an *effect* is - from my knowledge - justified only after a very thorough theoretical analysis. In practice, this results in concepts not being defined very clearly (e.g. flux imbalance, pore-centric theory, electroosmotic pumping - especially what is strong electroosmotic pumping?, etc) which makes reading difficult.

Furthermore the explanations regarding the Finite Element Analysis in Sec 5 are from my point not sufficient. In my experience, a high resolution meshing in the double layer (Debye length!) and at the tip of the nanocapillary is very important to get reliable results. Smooth curvatures at the tip yield different results than pointy tips. The considered geometry was not presented in enough detail The applied boundary conditions as I read the section are not complete (BC of the concentrations?) and the NP equation seems not to be stated correctly (where is the convection term that is stated to be considered?). In total, I'd be happy to see more details on the methods to be able to be sure the applied methodology is sound.

In total that brings me to the conclusion that the presented paper shows very interesting aspects of an exciting system with great details but the analysis is not concise enough to be published on the level targeted at. From my own experience (not only published results, but also preliminary studies with FE analysis of capillary nanopores), I know that achieving concise understanding is not easy. I myself have not managed to do so and encourage the authors to continue this **endeavor.**

REVIEWER COMMENTS

Reviewer #1 (Remarks to the Author):

The authors have addressed all points raised in the previous round of review.

Although the manuscript can be accepted as is, the authors are asked to consider a replacement to their frequently used language construct "to pump ions into a nanopore". To this reviewer, the construct implies that ion concentration continues to increase, as in "to pump water into a bucket". Perhaps an easy fix would be to replace "into" with "through", which does not imply accumulation.

Congratulations are in order for the authors on completing such a novel and exciting study

Reviewer #2 (Remarks to the Author):

I am satisfied that the authors have addressed all my concerns raised in connection with the previous version of the manuscript.

Reviewer #3 (Remarks to the Author):

I'd like to thank the authors for assessing my and the other reviewers' criticism and substantially improving the quality of the manuscript. This makes many experiments, parameters, concepts and ideas more clear. While the experimental part of the manuscript still is very impressive and especially the finding that identical asymmetric salt conditions, the sign of the observed events inverts when at the same time inverting the voltage and the DNA concentration.

However I my arguments regarding the weakness of the theoretical part could not yet be ruled out. I think the complexity of the system requires a rigorous analysis and a very careful handling of the wide range of theoretical concepts. In the theoretical analysis of electrokinetic phenomena, typically a linear expansion in the applied driving forces is performed (e.g. O'Brian & Wight Electrophoretic Mobility of a Spherical Colloidal Particle, 1978). Going beyond the linearity assumption is a strong but required step for understanding e.g. conduction rectification of nanopores. All effects that are not captured in the linear response require an especially thorough consideration and many intuitions and concepts from the "classical" theory are not valid any more. One example is the notion of conductance - the concept is based on the linear response to infinitesimal perturbation (=linear response theory, with a very sound apparatus of StatMech supporting this concept, Fluctuation-Dissipation theorem, reciprocal relations, etc). I don't believe that taking the slope of only the positive branch of the I-V curve is adequate. Flux imbalance in a system with weak boundary effects due to small Debye length clearly is a nonlinear phenomenon. It can only occur as an effect of strong applied driving forces. Being itself a result of strong nonlinearity, using the concept of flux imbalance as a *cause* rather than an *effect* is - from my knowledge - justified only after a very thorough theoretical analysis.

Response: We thank the reviewer for the insightful discussion included above. The terminology used by the reviewer was difficult to understand in the beginning; for example, what was meant by “non-linear response”. In the end, reviewer has led us to review the theoretical underpinnings in greater depth and have come to the same conclusion that, yes, classical theory is not enough to describe the phenomenon. The most pertinent topic, which we believe the reviewer was implying about here, was the concept of net neutrality and its violations both theoretically and experimentally. Net neutrality violations are rarely studied in the literature especially at the nanoscale and are a common assumption in Nernst-Planck equations. It is a topic that is also critical to flux imbalance. For example, flux imbalance is the cause of violating net neutrality. In order to NOT assume net neutrality, a full coupling between electrostatics and ionic concentrations was performed. This is precisely the reason that we observed voltage changes due to ionic concentrations (the charge density was defined using Poisson’s equation). Thus net neutrality is not a requirement for Poisson-Nernst-Planck equations (as used here). Indeed, non-linear responses are characteristic of violations of net neutrality and fully support this reviewer’s comments. I believe that the model has taken into account these non-linear phenomena: for example as the flux imbalance changes the concentration of ions, the voltage at the pore would be altered, and the flux of ions would be altered in response. Therefore, a graph of flux as a function of time should not be linear as the reviewer suggested. We investigated this using a time-dependent model and indeed found that the flux (and intra-pore electric field) decays on the order of nano-seconds to a steady state. A discussion of net neutrality and its role in assessing numerical models will be added to the manuscript. We have also included the following citation:

“Breakdown of electroneutrality in nanopores”-- <https://doi.org/10.1016/j.jcis.2020.05.109>

Manuscript Changes:

The following simulation result was included in the SI showing the dynamic and coupled relationship between ion flux and the electric field at the nanopore):

Intro Text:

In summary of our findings, DNA CEs are extremely cation-, pore size-, and voltage-specific and potentially the result of an imbalance of ionic fluxes and leads to charge density polarization and a violation of net neutrality²⁸. We utilize a Poisson-Nernst-Planck (PNP) model to describe

the flux imbalance between cation and anions within a nanopore which differs from the more traditional Nernst-Planck (NP) equations in how electro-neutrality and charge conservation is formulated. The PNP model more accurately describes the boundary layers (1-10 nm) at electrodes and charged surfaces²⁸. For nanopores that are on the same order of magnitude as the boundary layers, the PNP equations are a more complete treatment of charged species transport. The net effect is that flux imbalances have the ability to change the space charge density and the voltage throughout the fluidic system.

Results Text:

We have proposed a pore-centric model of CEs that is based on the dynamic distribution of ions inside of the nanopore. Volume exclusion is the typical mechanism of observing REs and we believe volume exclusion is still the main mechanism of CEs as well; both yield a transient ionic perturbation based on molecular occupancy of the pore. Since the voltage at the extreme ends of the fluidic reservoirs is clamped, charge build-up (i.e., potassium) tends to generate a voltage that, in turn, lowers the effective voltage for ion conduction at the pore. Inherent to a system with cation/anion flux imbalances is the concept of net neutrality, which is, by definition, violated by the conditions discussed here. Since electrostatics and ionic concentration profiles are coupled, voltage and ion flow are linked mechanistically. That is, especially with low electrolyte conditions, excess of either ion (cation or anion) could increase or decrease the voltage drop through the tapered region. The model developed for this study avoided the use of classical Nernst-Planck equations which assume net neutrality. Instead, a Poisson-Nernst-Planck (PNP) model was developed which permits ionic modulation of the electrostatic system.

In practice, this results in concepts not being defined very clearly (e.g. flux imbalance, pore-centric theory, electroosmotic pumping - especially what is strong electroosmotic pumping?, etc) which makes reading difficult.

Response: I believe we addressed the vague discussion of “flux imbalance” in the previous revision. EOF pumping terminology was extremely limited and defined specifically when used in the newest revisions. Electroosmotic pumping was more clearly defined in this version of the manuscript: the term represents the convective flow component of ionic transport and is delineated from other forms of pumping which drives ions or fluid flow. It also implies that energy is required to maintain the state. Since convection simply sets the frame of reference for electrophoretic motion, convective flows will always bias ion transport in favor of either anion or cation.

Manuscript Changes:

“Results from these studies predict strong electroosmotic pumping plays a role in driving DNA events and generating conductive events due to polarization effects (i.e. a pore-centric theory).”
Changed to....

“Results from these studies predict electroosmosis plays a role in driving DNA events and generating conductive events due to polarization effects (i.e. a pore-centric theory).”

Additional explanation of why pumping terminology was used: The terms EOF- and EPF-pumping are used here to signify that ions are being moved by the insertion of electrical energy and energy is required to maintain the system in that state.

Furthermore the explanations regarding the Finite Element Analysis in Sec 5 are from my point not sufficient. In my experience, a high resolution meshing in the double layer (Debye length!) and at the tip of the nanocapillary is very important to get reliable results. Smooth curvatures at the tip yield different results than pointy tips. The considered geometry was not presented in enough detail The applied boundary conditions as I read the section are not complete (BC of the concentrations?) and the NP equation seems not to be stated correctly (where is the convection term that is stated to be considered?). In total, I'd be happy to see more details on the methods to be able to be sure the applied methodology is sound.

Response: Thank you for the suggestions. Indeed, a greater level of detail is needed considering that NP equations alone are not sufficient (PNP equations are needed). More information is provided the manuscript as well as the SI.

Manuscript Changes:

Double layer was changed to debye length, and defined specifically in the revised text. More details of the model were also provided regarding the meshing and geometry. We also revised the total flux equation to include the convective term, which was mistakenly excluded or removed during revisions.

Section 5 revisions:

Finite element modelling was developed using COMSOL Multiphysics. The nanopores geometries were built on the TEM images and pulling protocols achieved from the experimental studies. The pore diameters used in figures and text refer strictly to the internal diameters of the most constricting portion of the geometry. The corners of the pore were also curved with an arc radius of 4 nm to avoid anomalies typical of sharp corners. The meshing of the geometry was performed with boundary layers accentuated for increased resolution and accuracy of the ionic flux within these layers (1-10 nm from all surfaces). A conical nanopore with a 25 nm diameter pore and a 4° half-cone angle was used unless otherwise stated. The diffusion coefficients were considered 2E-9 [m²/s] and 1.78E-9 [m²/s] for the potassium (K⁺) and chloride (Cl⁻) ions, respectively. The Poisson, Nernst-Planck, and Navier-Stokes equations were simultaneously solved to model the ionic behavior in a 2D axisymmetric steady-state model. The Poisson's equation [$\nabla^2(V) = -\rho/\epsilon$] described the relationship between the electric potential and ion transport mechanism. An important dimensional quantity in the Poisson equation is the Debye length, defined as $x_D = \sqrt{\frac{RT\epsilon_0\epsilon_s}{F^2I}}$ where I is the ionic strength and F is Faraday's constant. Poissons equation can also be written as $\rho = F \sum_i c_i z_i$ where ρ is the space charge density of the

fluidic domain. In COMSOL, the space charge density was specified within the *Electrostatics* module as well as the *Transport of Diluted Species* module as a volume force acting on the fluid (electroosmotic flow) and was defined specifically for binary electrolytes as: $\rho = N \cdot e \cdot z_1 \cdot c_1 + N \cdot e \cdot z_2 \cdot c_2$ where for the electrolyte containing c_1 (K^+) and c_2 (Cl^-) ionic species, z and e were set as the valency and electron charge, respectively. The electrostatics boundary condition used for the glass was set at a surface charge density of $-2E-2$ [C/m^2] in the vicinity of the pore opening to consider the surface charge contributions. The electric potential was set as variable field and the initial values were defined as zero potential.

The Poisson-Nernst-Planck equation was solved for the transport properties and ionic fluxes using convection, diffusion, and migration terms. The equation was described as: $J_i = -D_i \nabla c_i - z_i \mu_{m,i} F c_i \nabla^2 V + c_i u$ where J_i , D_i , c_i , u , $\mu_{m,i}$ and z_i are the ion flux, ion diffusion coefficient, concentration, fluid velocity, ion mobility and the charge number respectively. A no flux ($J=0$) condition was defined on the nanopore walls. The initial concentrations values of K^+ and Cl^- species were set to $10E-3$ [mol/L] for the entire domain. The inlet and outlet of the fluidic chamber was defined using an open boundary condition and the concentrations of the bulk electrolyte. The open boundary condition allows for convective inflow and outflow to occur which is important since convection currents could affect the concentration of ions within the system. The concentration of any “inflow” is set at the bulk value of the concentration. The fluid flow and pressure were modeled by the Navier-Stokes’s equation as: $\rho(u \cdot \nabla)u = (-\nabla_p + \eta \nabla^2 u - F(\sum z_i c_i) \nabla \Phi$. The u and Φ are the position dependent velocity field and potential field, z_i and c_i are species i charge and concentration in solution, ρ and η are the fluid density and dynamic viscosity, p is the pressure and F is the Faraday’s constant. Initial values of zero were assigned to the velocity field and pressure. The boundary condition for the pore wall was set to be $u=0$ (no-slip). To model the fluid flow, the volumetric force on the fluid was defined as ions space charge density multiplied by the electric field vectors. The fluid velocity was averaged over a 2D line spanning the nanopore orifice width. Peak velocity was at the center of the pore due to the zero potential boundary condition for the pore walls. Velocity and pressure were specified as the boundary conditions for the inlet and outlet, respectively.

In total that brings me to the conclusion that the presented paper shows very interesting aspects of an exciting system with great details but the analysis is not concise enough to be published on the level targeted at. From my own experience (not only published results, but also preliminary studies with FE analysis of capillary nanopores), I know that achieving concise understanding is not easy. I myself have not managed to do so and encourage the authors to continue this endeavor.

Response: We greatly appreciate the discussion and realize that net neutrality is by far the most prevailing theory and nearly assumed by default in most experiments. It is clearly not the case here: in both asymmetric salt conditions and pores that have convection (EOF or pressure-induced). Both conditions bias the transport towards that of cations or anions.

Manuscript changes: None, but we are willing to revise the manuscript if this reviewer has any specific requests (or concerns).

Peer review comments, third round review –

Reviewer #3 (Remarks to the Author):

I am grateful for the authors' patience with my comments and criticism. I myself will consider the concept of net neutrality and would like to learn more about it. I believe the manuscript changes make the manuscript more insightful and I would like to thank the authors for following my thoughts.

Congratulations for such an impressive work.

RESPONSE TO REVIEWERS

Reviewer #3 (Remarks to the Author):

I am grateful for the authors' patience with my comments and criticism. I myself will consider the concept of net neutrality and would like to learn more about it. I believe the manuscript changes make the manuscript more insightful and I would like to thank the authors for following my thoughts.

Congratulations for such an impressive work.

Response to Reviewer: We are extremely appreciative for the feedback to make this manuscript stronger and more insightful. Thank you for taking the time to review our documents!